# The Benefits of Being Distributional: Small-Loss Bounds for Reinforcement Learning

**Kaiwen Wang**    **Kevin Zhou**    **Runzhe Wu**    **Nathan Kallus**    **Wen Sun**
Cornell University
{kw437,klz23,rw646,kallus,ws455}@cornell.edu

## Abstract

While distributional reinforcement learning (DistRL) has been empirically effective, the question of when and why it is better than vanilla, non-distributional RL has remained unanswered. This paper explains the benefits of DistRL through the lens of small-loss bounds, which are instance-dependent bounds that scale with optimal achievable cost. Particularly, our bounds converge much faster than those from non-distributional approaches if the optimal cost is small. As warmup, we propose a distributional contextual bandit (DistCB) algorithm, which we show enjoys small-loss regret bounds and empirically outperforms the state-of-the-art on three real-world tasks. In online RL, we propose a DistRL algorithm that constructs confidence sets using maximum likelihood estimation. We prove that our algorithm enjoys novel small-loss PAC bounds in low-rank MDPs. As part of our analysis, we introduce the $\ell_1$ distributional eluder dimension which may be of independent interest. Then, in offline RL, we show that pessimistic DistRL enjoys small-loss PAC bounds that are novel to the offline setting and are more robust to bad single-policy coverage.

## 1   Introduction

The goal of reinforcement learning (RL) is to learn a policy that minimizes/maximizes the mean loss/return (*i.e.*, cumulative costs/rewards) along its trajectory. Classical approaches, such as $Q$-learning [Mnih et al., 2015] and policy gradients [Kakade, 2001], often learn $Q$-functions via least square regression, which represent the mean loss-to-go and act greedily with respect to these estimates. By Bellman's equation, $Q$-functions suffice for optimal decision-making and indeed these approaches have vanishing regret bounds, suggesting we only need to learn means well [Sutton and Barto, 2018]. Since the seminal work of Bellemare et al. [2017], however, numerous developments showed that learning the *whole* loss distribution can actually yield state-of-the-art performance in stratospheric balloon navigation [Bellemare et al., 2020], robotic grasping [Bodnar et al., 2020], algorithm discovery [Fawzi et al., 2022] and game playing benchmarks [Hessel et al., 2018, Dabney et al., 2018a, Barth-Maron et al., 2018]. In both online [Yang et al., 2019] and offline RL [Ma et al., 2021], distributional RL (DistRL) algorithms often perform better and use fewer samples in challenging tasks when compared to standard approaches that directly estimate the mean.

Despite learning the whole loss distribution, DistRL algorithms use only the mean of the learned distribution for decision making, not extracting any additional information such as higher moments. In other words, DistRL is simply employing a different and seemingly roundabout way of learning the mean: first, learn the loss-to-go distribution via distributional Bellman equations, and then, compute the mean of the learned distribution. Lyle et al. [2019] provided some empirical explanations of the benefits of this two-step approach, showing that learning the distribution, *e.g.*, its moments or quantiles, is an auxiliary task that leads to better representation learning. However, the theoretical

question remains: does DistRL, *i.e.*, learning the distribution and then computing the mean, yield provably stronger finite-sample guarantees and if so stronger how and when?

In this paper, we provide the first mathematical basis for the benefits of DistRL via the lens of small-loss bounds, which are instance-dependent bounds that depend on the minimum achievable cost in the problem [Agarwal et al., 2017].[1] For example in linear MDPs, typical worst-case regret bounds scale on the order of $\text{poly}(d, H)\sqrt{K}$, where $d$ is the feature dimension, $H$ is the horizon, and $K$ is the number of episodes [Jin et al., 2020b]. In contrast, small-loss bounds will scale on the order of $\text{poly}(d, H)\sqrt{K \cdot V^\star} + \text{poly}(d, H)\log(K)$, where $V^\star = \min_\pi V^\pi$ is the optimal expected cumulative cost for the problem. We assume cumulative costs are normalized in $[0, 1]$ without loss of generality. As $V^\star$ becomes negligible (approaches 0), the first term vanishes and the small-loss bound yields a faster convergence rate of $\mathcal{O}(\text{poly}(d, H)\log(K))$, compared to the $\mathcal{O}(\text{poly}(d, H)\sqrt{K})$ rate in standard uniform bounds. Since we always have $V^\star \leq 1$, small-loss bounds simply match the standard uniform bounds in the worst case.

As warm-up, we show that maximum likelihood estimation (MLE), *i.e.*, maximizing log-likelihood, can be used to obtain small-loss regret bounds for contextual bandits (CB), *i.e.*, the one-step RL setting. Then, we turn to the online RL setting, and propose an optimistic DistRL algorithm that optimizes over confidence sets constructed via MLE applied to the distributional Bellman equations. We prove our algorithm attains the first small-loss PAC bounds in low-rank MDPs [Agarwal et al., 2020]. Our proof uses a novel regret decomposition with triangular discrimination and also introduces the $\ell_1$ distributional eluder dimension, which generalizes the $\ell_2$ distributional eluder dimension of Jin et al. [2021a] and may be of independent interest. Furthermore, we design an offline distributional RL algorithm using the principle of pessimism, and show our algorithm obtains the first small-loss bounds in offline RL. Our offline small-loss bound holds under the weak single-policy coverage. Notably, our result has a novel robustness property that allows our algorithm to strongly compete with policies that either are well-covered or have small-loss, while prior approaches solely depended on the former. Finally, we find that our distributional CB algorithm empirically outperforms existing approaches in three challenging CB tasks.

Our key contributions are as follows:

1. As warm-up, we propose a distributional CB algorithm and prove that it obtains a small-loss regret bound (Section 4). We empirically demonstrate it outperforms state-of-the-art CB algorithms in three challenging benchmark tasks (Section 7).

2. We propose a distributional online RL algorithm that enjoys small-loss bounds in settings with low $\ell_1$ distributional eluder dimension, which we show can always capture low-rank MDPs. The $\ell_1$ distributional eluder dimension may be of independent interest (Section 5).

3. We propose a distributional offline RL algorithm and prove that it obtains the first small-loss bounds in the offline setting. Our small-loss guarantee exhibits a novel robustness to bad coverage, which implies strong improvement over more policies than existing results in the literature (Section 6).

In sum, we show that DistRL can yield small-loss bounds in both online and offline RL, which provide a concrete theoretical justification for the benefits of distribution learning in decision making.

## 2   Related Works

**Theory of Distributional RL**   Rowland et al. [2018, 2023] proved asymptotic convergence guarantees of popular distributional RL algorithms such as C51 [Bellemare et al., 2017] and QR-DQN [Dabney et al., 2018b]. However, these asymptotic results do not explain the *benefits* of distributional RL over standard approaches, since they do not imply stronger finite-sample guarantees than those obtainable with non-distributional algorithms. In contrast, our work shows that distributional RL yields adaptive finite-sample bounds that converge faster when the optimal cost of the problem is small. Wu et al. [2023] recently derived finite-sample bounds for distributional off-policy evaluation with MLE, while our offline RL section focuses on off-policy optimization.

---

[1]"First-order" generally refers to bounds that scale with the optimal value, either the maximum reward or the minimum cost. To highlight that we are minimizing cost, we call our bounds "small-loss".

**First-order bounds in bandits**   When maximizing rewards, first-order "small-return" bounds can be easily derived from EXP4 [Auer et al., 2002], since receiving the worst reward 0 with probability (w.p.) $\delta$ contributes at most $R^\star \delta$ to the regret[2]. When minimizing costs, receiving the worst loss 1 w.p. $\delta$ may induce large regret relative to $L^\star$ if $L^\star$ is small. To illustrate, if $R^\star = 0$ then all policies are optimal, so no learning is needed and the small-return bound is vacuous. Yet if $L^\star = 0$, sub-optimal policies may have a large gap from $L^\star$, so small-loss bounds in this regime are meaningful. Small-loss bounds are achievable in multi-arm bandits [Foster et al., 2016], semi-bandits [Neu, 2015, Lykouris et al., 2022], and CBs [Allen-Zhu et al., 2018, Foster and Krishnamurthy, 2021].

**First-order bounds in RL**   Jin et al. [2020a], Wagenmaker et al. [2022] obtained small-return regret for tabular and linear MDPs via concentration bounds that scale with the variance. The idea is that the return's variance is bounded by some multiple of the expected value, which is bounded by $V^\star$ in the reward-maximizing setting, *i.e.*, $\mathrm{Var}(\sum_h r_h \mid \pi^k) \leq c \cdot V^{\pi^k} \leq c \cdot V^\star$. However, the last inequality fails in the loss-minimizing setting, so the variance approach does not easily yield small-loss bounds. Small-loss regret for tabular MDPs was resolved by Lee et al. [2020, Theorem 4.1] using online mirror descent with the log-barrier on the occupancy measure. Moreover, Kakade et al. [2020, Theorem 3.8] obtains small-loss regret for linear-quadratic regulators (LQRs), but their Assumption 3 posits that the coefficient of variation for the cumulative costs is bounded, which is false in general even in tabular MDPs. To the best of our knowledge, there are no known first-order bounds for low-rank MDPs or in offline RL.

**Risk-sensitive RL**   A well-motivated use-case of DistRL is risk-sensitive RL, where the goal is to learn risk-sensitive policies that optimize some risk measure, *e.g.*, Conditional Value-at-Risk (CVaR), of the loss [Dabney et al., 2018b]. Orthogonal to risk-sensitive RL, this work focuses on the benefits of DistRL for standard risk-neutral RL. Our insights may lead to first-order bounds for risk-sensitive RL, which we leave as future work.

## 3   Preliminaries

As warmup, we begin with the contextual bandit problem with an arbitrary context space $\mathcal{X}$, finite action space $\mathcal{A}$ with size $A$ and conditional cost distributions $C : \mathcal{X} \times \mathcal{A} \to \Delta([0, 1])$. Throughout, we fix some dominating measure $\lambda$ on $[0, 1]$ (*e.g.*, Lebesgue for continuous or counting for discrete) and let $\Delta([0, 1])$ be all distributions on $[0, 1]$ that are absolutely continuous with respect to $\lambda$. We identify such a distribution with its density with respect to $\lambda$, and we also write $C(y \mid x, a)$ for $(C(x, a))(y)$. Let $K$ denote the number of episodes. At each episode $k \in [K]$, the learner observes a context $x_k \in \mathcal{X}$, samples an action $a_k \in \mathcal{A}$, and then receives a cost $c_t \sim C(x_t, a_t)$, which we assume to be normalized, *i.e.*, $c_t \in [0, 1]$. The goal is to design a learner that attains low regret with high probability, where regret is defined as

$$\mathrm{Regret}_{\mathrm{CB}}(K) = \sum_{k=1}^K \bar{C}(x_k, a_k) - \bar{C}(x_k, \pi^\star(x_k)),$$

where $\bar{f} = \int y f(y) \mathrm{d}\lambda(y)$ for any $f \in \Delta([0, 1])$ and $\pi^\star(x_k) = \arg\min_{a \in \mathcal{A}} \bar{C}(x_k, a)$.

The focus of this paper is reinforcement learning (RL) under the Markov Decision Process (MDP) model, with observation space $\mathcal{X}$, finite action space $\mathcal{A}$ with size $A$, horizon $H$, transition kernels $P_h : \mathcal{X} \times \mathcal{A} \to \Delta(\mathcal{X})$ and *cost* distributions $C_h : \mathcal{X} \times \mathcal{A} \to \Delta([0, 1])$ at each step $h \in [H]$. We start with the *Online RL* setting, which proceeds over $K$ episodes as follows: at each episode $k \in [K]$, the learner plays a policy $\pi^k \in [\mathcal{X} \to \Delta(\mathcal{A})]^H$; we start from a fixed initial state $x_1$; then for each $h = 1, 2, \ldots, H$, the policy samples an action $a_h \sim \pi_h^k(x_h)$, receives a cost $c_h \sim C_h(x_h, a_h)$, and transitions to the next state $x_{h+1} \sim P_h(x_h, a_h)$. Our goal is to compete with the optimal policy that minimizes expected the loss, *i.e.*, $\pi^\star \in \arg\min_{\pi \in \Pi} V^\pi$ where $V^\pi = \mathbb{E}_\pi\left[\sum_{h=1}^H c_h\right]$. Regret bounds aim to control the learner's regret with high probability, where regret is defined as,

$$\mathrm{Regret}_{\mathrm{RL}}(K) = \sum_{k=1}^K V^{\pi^k} - V^\star.$$

If the algorithm returns a single policy $\widehat{\pi}$, it is desirable to obtain a Probably Approximately Correct (PAC) bound on the sub-optimality of $\widehat{\pi}$, *i.e.*, $V^{\widehat{\pi}} - V^\star$.

---

[2]Assume rewards/losses in $[0, 1]$ and $R^\star / L^\star$ is the maximum/minimum expected reward/loss.

The third setting we study is *Offline RL*, where instead of needing to actively explore and collect data ourselves, we are given $H$ datasets $\mathcal{D}_1, \mathcal{D}_2, \ldots, \mathcal{D}_H$ to learn a good policy $\widehat{\pi}$. Each $\mathcal{D}_h$ contains $N$ *i.i.d.* samples $(x_{h,i}, a_{h,i}, c_{h,i}, x'_{h,i})$ from the process $(x_{h,i}, a_{h,i}) \sim \nu_h, c_{h,i} \sim C_h(x_{h,i}, a_{h,i}), x'_{h,i} \sim P_h(x_{h,i}, a_{h,i})$, where $\nu_h \in \Delta(\mathcal{X} \times \mathcal{A})$ is arbitrary, *e.g.*, the visitations of many policies from the current production system. The goal is to design an offline procedure with a PAC guarantee for $\widehat{\pi}$, which should improve over the data generating process.

**Distributional RL**   For a policy $\pi$ and $h \in [H]$, let $Z_h^\pi(x_h, a_h) \in \Delta([0,1])$ denote the distribution of the loss-to-go $\sum_{t=h}^H c_t$ conditioned on rolling in $\pi$ from $x_h, a_h$. The expectation of the above is $Q_h^\pi(x_h, a_h) = \bar{Z}_h^\pi(x_h, a_h)$ and $V_h^\pi(x_h) = \mathbb{E}_{a_h \sim \pi_h(x_h)}[Q_h^\pi(x_h, a_h)]$. We use $Z_h^\star, Q_h^\star, V_h^\star$ to denote these quantities with $\pi^\star$. Recall the regular Bellman operator acts on a function $f : \mathcal{X} \times \mathcal{A} \to [0,1]$ as follows: $\mathcal{T}_h^\pi f(x, a) = \bar{C}_h(x, a) + \mathbb{E}_{x' \sim P_h(x,a), a' \sim \pi(x')}[f(x', a')]$. Analogously, the distributional Bellman operator [Morimura et al., 2012, Bellemare et al., 2017] acts on a conditional distribution $d : \mathcal{X} \times \mathcal{A} \to \Delta([0,1])$ as follows: $\mathcal{T}_h^{\pi,D} d(x, a) \overset{D}{=} C_h(x, a) + d(x', a')$, where $x' \sim P_h(x, a), a' \sim \pi(x')$ and $\overset{D}{=}$ denotes equality of distributions. Another way to think about the distributional Bellman operator is that a sample $z \sim \mathcal{T}_h^{\pi,D} d(x, a)$ is generated as follow: $z := c + y$, where $c \sim C_h(x, a), x' \sim P_h(x, a), a' \sim \pi(x'), y \sim d(x', a')$. We will also use the Bellman optimality operator $\mathcal{T}_h^\star$ and its distributional variant $\mathcal{T}_h^{\star,D}$, defined as follows: $\mathcal{T}_h^\star f(x, a) = \bar{C}_h(x, a) + \mathbb{E}_{x' \sim P_h(x,a)}[\min_{a \in \mathcal{A}} f(x', a')]$ and $\mathcal{T}_h^{\star,D} d(x, a) \overset{D}{=} C_h(x, a) + d(x', a')$ where $x' \sim P_h(x, a), a' = \arg\min_a \bar{d}(x', a)$. Please see Table 2 for an index of notations.

## 4   Warm up: Small-Loss Regret for Distributional Contextual Bandits

In this section, we propose an efficient reduction from CB to online maximum likelihood estimation (MLE), which is the standard tool for distribution learning that we will use throughout the paper. In our CB algorithm, we balance exploration and exploitation with the reweighted inverse gap weighting (ReIGW) of Foster and Krishnamurthy [2021], which defines a distribution over actions given predictions $\widehat{f} \in \mathbb{R}^A$ and a parameter $\gamma \in \mathbb{R}_{++}$: setting $b = \arg\min_{a \in \mathcal{A}} \widehat{f}(a)$ as the best action with respect to the predictions, the weight for any other action $a \neq b$ is,

$$\text{ReIGW}_\gamma(\widehat{f}, \gamma)[a] := \frac{\widehat{f}(b)}{A\widehat{f}(b) + \gamma(\widehat{f}(a) - \widehat{f}(b))}, \tag{1}$$

and the rest of the weight is allocated to $b$: $\text{ReIGW}_\gamma(\widehat{f}, \gamma)[b] = 1 - \sum_{a \neq b} \text{ReIGW}_\gamma(\widehat{f}, \gamma)[a]$.

---
**Algorithm 1** Distributional CB (DISTCB)
---
1: **Input:** number of episodes $K$, failure probability $\delta$, ReIGW learning rate $\gamma$.
2: Initialize any cost distribution $f^{(1)}$.
3: **for** episode $k = 1, 2, \ldots, K$ **do**
4:     Observe context $x_k$.
5:     Sample action $a_k \sim p_k = \text{ReIGW}(\bar{f}^{(k)}(x_k, \cdot), \gamma)$ from Eq. (1).
6:     Observe cost $c_k \sim C(x_k, a_k)$ and update online MLE oracle with $((x_k, a_k), c_k)$.
7: **end for**
---

We propose **Dist**ributional **C**ontextual **B**andit (DISTCB) in Algorithm 1, a two-step procedure for each episode $k \in [K]$. Upon seeing context $x_k$, DISTCB first samples an action $a_k$ from ReIGW generated by means of our estimated cost distributions for each action, *i.e.*, $\widehat{f}(a) = \bar{f}^{(k)}(x_k, a), \forall a \in \mathcal{A}$ (Line 5). Then, DISTCB updates $f^{(k)}(\cdot \mid x_k, a_k)$ by maximizing the log-likelihood to estimate the conditional cost distribution $C(\cdot \mid x_k, a_k)$ (Line 6). Formally, this second step is achieved via an online MLE oracle with a realizable distribution class $\mathcal{F}_{CB} \subset \mathcal{X} \times \mathcal{A} \to \Delta([0,1])$; let $\text{Regret}_{\log}(K)$ be some upper bound on the log-likelihood regret for all possibly adaptive sequences $\{x_k, a_k, c_k\}_{k \in [K]}$,

$$\sum_{k=1}^K \log C(c_k \mid x_k, a_k) - \log f^{(k)}(c_k \mid x_k, a_k) \leq \text{Regret}_{\log}(K).$$

Under *realizability*, $C \in \mathcal{F}_{CB}$, we expect $\text{Regret}_{\log}(K) \in \mathcal{O}(\log(K))$. For instance, if $\mathcal{F}_{CB}$ is finite, exponentially weighted average forecaster guarantees $\text{Regret}_{\log}(K) \leq \log|\mathcal{F}_{CB}|$ [Cesa-Bianchi and Lugosi, 2006, Chapter 9]. We now state our main result for DISTCB.

**Theorem 4.1.** *For any $\delta \in (0,1)$, w.p. at least $1-\delta$, running* DISTCB *with $\gamma = 10A \vee \sqrt{\frac{40A(C^\star + \log(1/\delta))}{112(\mathrm{Regret}_{\log}(K) + \log(1/\delta))}}$ has regret scaling with $C^\star = \sum_{k=1}^{K} \min_{a \in \mathcal{A}} \bar{C}(x_k, a)$,*

$$\mathrm{Regret}_{\mathrm{DISTCB}}(K) \leq 232\sqrt{AC^\star \, \mathrm{Regret}_{\log}(K) \log(1/\delta)} + 2300A\big(\mathrm{Regret}_{\log}(K) + \log(1/\delta)\big).$$

The dominant term scales with the optimal sum of costs $\sqrt{C^\star}$ which shows that DISTCB obtains small-loss regret. DISTCB is also computationally efficient since each episode simply requires computing the ReIGW. FastCB is the only other computationally efficient CB algorithm with small-loss regret [Foster and Krishnamurthy, 2021, Theorem 1]. Our bound matches that of FastCB in terms of dependence on $A, C^\star$ and $\log(1/\delta)$. Our key difference with FastCB is the online supervised learning oracle: in DISTCB, we aim to learn the conditional cost distribution by maximizing log-likelihood, while FastCB aims to perform regression with the binary cross-entropy loss. In Section 7, we find that DISTCB empirically outperforms SquareCB and FastCB in three challenging CB tasks, which reinforces the practical benefits of distribution learning in CB setting.

## 4.1 Proof Sketch

First, apply the per-round inequality for ReIGW [Foster and Krishnamurthy, 2021, Theorem 4] to get,

$$\mathrm{Regret}_{\mathrm{DistCB}}(K) \lesssim \sum_{k=1}^{K} \mathbb{E}_{a_k \sim p_k} \left[ \frac{A}{\gamma} \bar{C}(s_k, a_k) + \gamma \underbrace{\frac{\big(\bar{f}^{(k)}(s_k, a_k) - \bar{C}(s_k, a_k)\big)^2}{\bar{f}^{(k)}(s_k, a_k) + \bar{C}(s_k, a_k)}}_{\bigstar} \right].$$

For any distributions $f, g \in \Delta([0,1])$, their triangular discrimination[3] is defined as $D_{\triangle}(f \parallel g) := \int \frac{(f(y) - g(y))^2}{f(y) + g(y)} d\lambda(y)$. The key insight is that $\bigstar$ can be bounded by the triangular discrimination of $f^{(k)}(s_k, a_k)$ and $C(s_k, a_k)$: by Cauchy-Schwartz and $y^2 \leq y$ for $y \in [0,1]$, we have $\bar{f} - \bar{g} = \int y(f(y) - g(y)) d\lambda(y) \leq \sqrt{\int y(f(y) + g(y)) d\lambda(y)} \sqrt{\int \frac{(f(y) - g(y))^2}{f(y) + g(y)} d\lambda(y)}$, and hence,

$$\big|\bar{f} - \bar{g}\big| \leq \sqrt{(\bar{f} + \bar{g}) D_{\triangle}(f \parallel g)}. \tag{$\triangle_1$}$$

So, Eq. ($\triangle_1$) implies that $\bigstar$ is bounded by $D_{\triangle}(f^{(k)}(s_k, a_k) \parallel C(s_k, a_k))$. Since $D_{\triangle}$ is equivalent (up to universal constants) to the squared Hellinger distance, Foster et al. [2021, Lemma A.14] implies the above can be bounded by the online MLE regret, so w.p. at least $1-\delta$, we have

$$\mathrm{Regret}_{\mathrm{DistCB}}(K) \lesssim \sum_{k=1}^{K} \frac{A}{\gamma}\big(\bar{C}(s_k, a_k) + \log(1/\delta)\big) + \gamma\big(\mathrm{Regret}_{\log}(K) + \log(1/\delta)\big).$$

From here, we just need to rearrange terms and set the correct $\gamma$. Appendix C contains the full proof.

## 5 Small-Loss Bounds for Online Distributional RL

We now extend our insights to the online RL setting and propose a DistRL perspective on GOLF [Jin et al., 2021a]. While GOLF constructs confidence sets of near-minimizers of the squared Bellman error loss, we propose to construct these confidence sets using near-maximizers of the log-likelihood loss to approximate MLE. To leverage function approximation for learning conditional distributions, we use a generic function class $\mathcal{F} \subseteq (\mathcal{X} \times \mathcal{A} \to \Delta([0,1]))^H$ where each element $f \in \mathcal{F}$ is a tuple $f = (f_1, \ldots, f_H)$ such that each $f_h$ is a candidate estimator for $Z_h^\star$, the distribution of loss-to-go $\sum_{t=h}^{H} c_t$ under $\pi^\star$. For notation, $f_{H+1}(x, a) = \delta_0$ denotes the dirac at zero for all $x, a$.

We now present our **O**ptimistic **Dis**tributional **C**onfidence set **O**ptimization (O-DISCO) algorithm in Algorithm 2, consisting of three key steps per episode. At episode $k \in [K]$, O-DISCO first identifies the $f^{(k)}$ with the minimal expected value at $h = 1$ over the previous confidence set $\mathcal{F}_{k-1}$ (Line 4). This step induces *global optimism*. Then, O-DISCO collects data for this episode by rolling in with the greedy policy $\pi^k$ with respect to the mean of $f^{(k)}$ (Line 6). Finally, O-DISCO constructs a

---

[3]Triangular discrimination is also known as Vincze-Le Cam divergence [Vincze, 1981, Le Cam, 2012].

---

**Algorithm 2 O**ptimistic **Dist**ributional **C**onfidence set **O**ptimization (O-DISCO)

1: **Input:** number of episodes $K$, distribution class $\mathcal{F}$, threshold $\beta$.
2: Initialize $\mathcal{D}_{h,0} \leftarrow \emptyset$ for all $h \in [H]$, and set $\mathcal{F}_0 = \mathcal{F}$.
3: **for** episode $k = 1, 2, \ldots, K$ **do**
4:     Set optimistic estimate $f^{(k)} = \arg\min_{f \in \mathcal{F}_{k-1}} \min_a \bar{f}_1(x_1, a)$.
5:     Set $\pi_h^k(x) = \arg\min_a \bar{f}_h^{(k)}(x, a)$.
6:     Roll out $\pi^k$ and obtain a trajectory $x_{1,k}, a_{1,k}, c_{1,k}, \ldots, x_{H,k}, a_{H,k}, c_{H,k}$.
      For each $h \in [H]$, augment the dataset $\mathcal{D}_{h,k} = \mathcal{D}_{h,k-1} \cup \{(x_{h,k}, a_{h,k}, c_{h,k}, x_{h+1,k})\}$.
7:     For all $(h, f) \in [H] \times \mathcal{F}$, sample $y_{h,i}^f \sim f_{h+1}(x_{h,i}', a')$ and $a' = \arg\min_a \bar{f}_{h+1}(x_{h,i}', a)$,
      where $(x_{h,i}, a_{h,i}, c_{h,i}, x_{h,i}')$ is the $i$-th datapoint of $\mathcal{D}_{h,k}$. Then, set $z_{h,i}^f = c_{h,i} + y_{h,i}^f$ and
      define the confidence set

$$\mathcal{F}_k = \left\{ f \in \mathcal{F} : \sum_{i=1}^k \log f_h(z_{h,i}^f \mid x_{h,i}, a_{h,i}) \geq \max_{g \in \mathcal{F}_h} \sum_{i=1}^k \log g(z_{h,i}^f \mid x_{h,i}, a_{h,i}) - 7\beta, \forall h \in [H] \right\}.$$

8: **end for**
9: **Output:** $\bar{\pi} = \text{unif}(\pi^{1:K})$.

---

confidence set $\mathcal{F}_k$ by including a function $f$ if it exceeds a threshold on the log-likelihood objective using data $z_{h,i}^f \sim \mathcal{T}_h^{\star,D} f_{h+1}(x_{h,i}, a_{h,i})$ for all steps $h$ simultaneously (Line 7). This step is called *local fitting*, as each $f \in \mathcal{F}_k$ has the property that $f_h$ is close-in-distribution to $\mathcal{T}_h^{\star,D} f_{h+1}$ for all $h$. We highlight that O-DISCO only learns the distribution for estimating the mean, *i.e.*, Lines 4 and 6 only use the mean $\bar{f}$. This seemingly roundabout way of estimating the mean is exactly how distributional RL algorithms such as C51 differ from the classic DQN.

To ensure that MLE succeeds for the Temporal-Difference (TD) style confidence sets, we need the following distributional Bellman Completeness (BC) condition introduced in Wu et al. [2023].

**Assumption 5.1** (Bellman Completeness). *For all $\pi, h \in [H]$, $f_{h+1} \in \mathcal{F}_{h+1} \implies \mathcal{T}_h^{\pi,D} f_{h+1} \in \mathcal{F}_h$.*

### 5.1 The $\ell_1$ Distributional Eluder Dimension

We now introduce the $\ell_1$ distributional eluder dimension. Let $\mathcal{S}$ be an abstract input space, let $\Psi$ be a set of functions mapping $\mathcal{S} \to \mathbb{R}$ and let $\mathcal{D}$ be a set of distributions on $\mathcal{S}$.

**Definition 5.2** ($\ell_p$-distributional eluder dimension). *For any function class $\Psi \subseteq \mathcal{S} \to \mathbb{R}$, distribution class $\mathcal{D} \subseteq \Delta(\mathcal{S})$ and $\varepsilon > 0$, the $\ell_p$-distributional eluder dimension (denoted by $\text{DE}_p(\Psi, \mathcal{D}, \varepsilon)$) is the length $L$ of the longest sequence $d^{(1)}, d^{(2)}, \ldots, d^{(L)} \subseteq \mathcal{D}$ such that there exists $\varepsilon' \geq \varepsilon$, such that for all $t \in [L]$, we have that there exists $f \in \Psi$ such that $|\mathbb{E}_{d^{(t)}} f| > \varepsilon$ and also $\sum_{i=1}^{t-1} |\mathbb{E}_{d^{(i)}} f|^p \leq \varepsilon^p$.*

When $p = 2$, this is exactly the $\ell_2$ distributional eluder of Jin et al. [2021a, Definition 7]. We're particularly interested in the $p = 1$ case, which can be used with MLE's generalization bounds. The following is a key pigeonhole principle for the $\ell_1$ distributional eluder dimension.

**Theorem 5.3.** *Let $C := \sup_{d \in \mathcal{D}, f \in \Psi} |\mathbb{E}_d f|$ be the envelope. Fix any $K \in \mathbb{N}$ and sequences $f^{(1)}, \ldots, f^{(K)} \subseteq \Psi$, $d^{(1)}, \ldots, d^{(K)} \subseteq \mathcal{D}$. Let $\beta$ be a constant such that for all $k \in [K]$, we have, $\sum_{i=1}^{k-1} |\mathbb{E}_{d^{(i)}} f^{(k)}| \leq \beta$. Then, for all $k \in [K]$, we have*

$$\sum_{t=1}^k \left| \mathbb{E}_{d^{(t)}} f^{(t)} \right| \leq \inf_{0 < \varepsilon \leq 1} \{ \text{DE}_1(\Psi, \mathcal{D}, \varepsilon)(2C + \beta \log(C/\varepsilon)) + k\varepsilon \}.$$

As we'll see later, Theorem 5.3 is the key tool that transfers triangular discrimination guarantees on the training distribution to any new test distribution. Another key property is that the $\ell_1$ dimension generalizes the original $\ell_2$ dimension of Jin et al. [2021a].

**Lemma 5.4.** *For any $\Psi, \mathcal{D}$ and $\varepsilon > 0$, we have $\text{DE}_1(\Psi, \mathcal{D}, \varepsilon) \leq \text{DE}_2(\Psi, \mathcal{D}, \varepsilon)$.*

Finally, we note that our distributional eluder dimension generalize the regular $\ell_1$ eluder from Liu et al. [2022], which can be seen by taking $\mathcal{D}$ to be dirac distributions.

## 5.2 Small-Loss Bounds for O-DISCO

We will soon prove small-loss regret bounds with the "Q-type" dimension, where "Q-type" refers to the fact that $\mathcal{S} = \mathcal{X} \times \mathcal{A}$. While low-rank MDPs are not captured by the "Q-type" dimension, they are captured by the "V-type" dimension where $\mathcal{S} = \mathcal{X}$ [Jin et al., 2021a, Du et al., 2021]. For PAC bounds with the V-type dimension, we need to slightly modify the data collection process in Line 6 with *uniform action exploration* (UAE). Instead of executing $\pi^k$ for a single trajectory, partially roll-out $\pi^k$ for $H$ times where for each $h \in [H]$, we collect $x_{h,k} \sim d_h^{\pi^k}$, take a random action $a_{h,k} \sim \text{unif}(\mathcal{A})$, observe $c_{h,k} \sim C_h(x_{h,k}, a_{h,k}), x'_{h,k} \sim P_h(x_{h,k}, a_{h,k})$ and augment the dataset $\mathcal{D}_{h,k} = \mathcal{D}_{h,k-1} \cup \{(x_{h,k}, a_{h,k}, c_{h,k}, x'_{h,k})\}$. The modified algorithm is detailed in Appendix B.

We lastly need to define the function and distribution classes measured by the distributional eluder dimension. The Q-type classes are $\mathcal{D}_h = \{(x,a) \mapsto d_h^\pi(x,a) : \pi \in \Pi\}$ and $\Psi_h = \{(x,a) \mapsto D_\triangle(f(x,a) \parallel \mathcal{T}^{\star,D} f(x,a)) : f \in \mathcal{F}\}$. Similarly, the V-type classes are $\mathcal{D}_{h,v} = \{x \mapsto d_h^\pi(x) : \pi \in \Pi\}$ and $\Phi_{h,v} = \{x \mapsto \mathbb{E}_{a \sim \text{Unif}(\mathcal{A})}[D_\triangle(f(x,a) \parallel \mathcal{T}^{\star,D} f(x,a))] : f \in \mathcal{F}\}$. Finally, define $\text{DE}_1(\varepsilon) = \max_h \text{DE}_1(\Psi_h, \mathcal{D}_h, \varepsilon)$ and $\text{DE}_{1,v}(\varepsilon) = \max_h \text{DE}_1(\Psi_{h,v}, \mathcal{D}_{h,v}, \varepsilon)$.

**Theorem 5.5.** *Suppose DistBC holds (Assumption 5.1). For any $\delta \in (0,1)$, w.p. at least $1 - \delta$, running O-DISCO with $\beta = \log(HK|\mathcal{F}|/\delta)$ guarantees the following regret bound,*

$$\text{Regret}_{\text{O-DISCO}}(K) \le 160 H \sqrt{K V^\star \text{DE}_1(1/K) \log(K) \beta} + 18000 H^2 \text{DE}_1(1/K) \log(K) \beta.$$

*If* UAE = TRUE *(Algorithm 4), then the learned mixture policy $\bar{\pi}$ is guaranteed to satisfy,*

$$V^{\bar{\pi}} - V^\star \le 160 H \sqrt{\frac{A V^\star \text{DE}_{1,v}(1/K) \log(K) \beta}{K}} + \frac{18000 H^2 A \text{DE}_{1,v}(1/K) \log(K) \beta}{K}.$$

Compared to prior bounds for GOLF [Jin et al., 2021a], the leading $\sqrt{K}$ terms in our bounds enjoy the same sharp dependence in $H, K$ and the eluder dimension. Our bounds further enjoy one key improvement: the leading terms are multiplied with the instance-dependent optimal cost $V^\star$, giving our bounds the *small-loss* property. For example, if $V^\star \le \mathcal{O}(1/\sqrt{K})$, then our regret bound converges at a fast $\mathcal{O}(H^2 \text{DE}_1(1/K) \log(K) \beta)$ rate. While there are existing first-order bounds in online RL, our bound significantly improves on their generality. For example, Zanette and Brunskill [2019], Jin et al. [2020a], Wagenmaker et al. [2022] used Bernstein bonuses that scale with the conditional variance and showed that careful analysis can lead to "small-return" bounds in tabular and linear MDPs. However, "small-return" bounds do not imply "small-loss" bounds and "small-loss" bounds are often harder to obtain[4]. While it is possible that surgical analysis with variance bonuses can lead to small-loss bounds in tabular and linear MDPs, this approach may not scale to settings with non-linear function approximation such as low-rank MDPs.

**On Bellman Completeness** Exponential error amplification can occur in online and offline RL under only realizability of $Q$ functions [Wang et al., 2021a,b,c, Foster et al., 2022]. With only realizability, basic algorithms such as TD and Fitted-$Q$-Evaluation (FQE) can diverge or converge to bad fixed point solutions [Tsitsiklis and Van Roy, 1996, Munos and Szepesvári, 2008, Kolter, 2011]. As a result, BC has risen as a *de facto* sufficient condition for sample efficient RL [Chang et al., 2022, Xie et al., 2021, Zanette et al., 2021]. Finally, we highlight that our method can be easily extended to hold under *generalized completeness*, *i.e.*, there exist function classes $\mathcal{G}_h$ such that $f_{h+1} \in \mathcal{F}_{h+1} \implies \mathcal{T}_h^{\pi,D} f_{h+1} \in \mathcal{G}_h$ [as in Jin et al., 2021a, Assumption 14]. Simply replace $\max_{g \in \mathcal{F}_h}$ in the confidence set construction with $\max_{g \in \mathcal{G}_h}$. While adding functions to $\mathcal{F}$ may break BC (as BC is not monotonic), we can always augment $\mathcal{G}$ to satisfy generalized completeness.

**Computational complexity** When taken as is, OLIVE [Jiang et al., 2017], GOLF, and our algorithms are version space methods that suffer from a computational drawback: optimizing over the confidence set is NP-hard [Dann et al., 2018]. However, the confidence set is purely for deep exploration via optimism and can be replaced by other computationally efficient exploration strategies. For example, $\varepsilon$-greedy suffices in problems that don't require deep and strategic exploration, *i.e.*, a large myopic exploration gap [Dann et al., 2022]. With $\varepsilon$-greedy, a replay buffer, and discretization, our algorithm essentially recovers C51 [Bellemare et al., 2017]. We leave developing and analyzing computationally efficient algorithms based on our insights as promising future work.

---

[4]In Appendix J, we show a slight modification of our approach also yields "small-return" bounds.

## 5.3 Instantiation with Low-Rank MDPs

The low-rank MDP [Agarwal et al., 2020] is a standard abstraction for non-linear function approximation used in theory [Uehara et al., 2021] and practice [Zhang et al., 2022, Chang et al., 2022].

**Definition 5.6** (Low-rank MDP). A transition model $P_h : \mathcal{X} \times \mathcal{A} \to \Delta(\mathcal{X})$ has rank $d$ if there exist unknown features $\phi_h^\star : \mathcal{X} \times \mathcal{A} \to \mathbb{R}^d, \mu_h^\star : \mathcal{X} \to \mathbb{R}^d$ such that $P_h(x' \mid x, a) = \phi_h^\star(x, a)^\top \mu_h^\star(x')$ for all $x, a, x'$. Also, assume $\max_{x,a} \|\phi_h^\star(x,a)\|_2 \leq 1$ and $\|\int g \mathrm{d}\mu_h^\star\|_2 \leq \|g\|_\infty \sqrt{d}$ for all functions $g : \mathcal{X} \to \mathbb{R}$. The MDP is called low-rank if $P_h$ is low-rank for all $h \in [H]$.

We now specialize Theorem 5.5 to low-rank MDPs with three key steps. First, we bound the V-type eluder dimension by $\mathrm{DE}_{1,v}(\varepsilon) \leq \mathcal{O}(d \log(d/\varepsilon))$, which is a known result that we reproduce in Theorem G.4. The next step requires access to a realizable $\Phi$ class, *i.e.*, for all $h \in [H]$, $\phi_h^\star \in \Phi$, which is a standard assumption for low-rank MDPs [Agarwal et al., 2020, Uehara et al., 2021, Mhammedi et al., 2023]. Given the realizable $\Phi$, we can construct a specialized $\mathcal{F}$ for the low-rank MDP: $\mathcal{F}^{\mathrm{lin}} = \mathcal{F}_1^{\mathrm{lin}} \times \cdots \times \mathcal{F}_H^{\mathrm{lin}} \times \mathcal{F}_{H+1}^{\mathrm{lin}}$ where $\mathcal{F}_{H+1}^{\mathrm{lin}} = \{\delta_0\}$ and for all $h \in [H]$,

$$\mathcal{F}_h^{\mathrm{lin}} = \left\{ f(z \mid x, a) = \langle \phi(x, a), w(z) \rangle \quad : \quad \phi \in \Phi, w : [0, 1] \to \mathbb{R}^d, \right. \tag{2}$$

$$\left. \text{s.t.} \ \max_z \|w(z)\|_2 \leq \alpha \sqrt{d} \ \text{ and } \ \max_{x,a,z} \langle \phi(x, a), w(z) \rangle \leq \alpha \right\},$$

where $\alpha := \max_{h,\pi,z,x,a} Z_h^\pi(z \mid x, a)$ is the largest mass for the cost-to-go distributions. In Appendix D, we show that $\mathcal{F}^{\mathrm{lin}}$ satisfies DistBC. Further, if costs are discretized into a uniform grid of $M$ points, its bracketing entropy is bounded by $\widetilde{\mathcal{O}}(dM + \log|\Phi|)$. Discretization is necessary to bound the statistical complexity of $\mathcal{F}^{\mathrm{lin}}$ and is common in practice, *e.g.*, C51 and Rainbow both set $M = 51$ which works well in Atari games [Bellemare et al., 2017, Hessel et al., 2018].

**Theorem 5.7.** *Suppose the MDP is low-rank. For any $\delta \in (0, 1)$, w.p. at least $1 - \delta$, running* O-DISCO *with* UAE=TRUE *and with $\mathcal{F}^{\mathrm{lin}}$ as described above learns a policy $\bar\pi$ such that,*

$$V^{\bar\pi} - V^\star \in \widetilde{\mathcal{O}}\left( H\sqrt{\frac{AdV^\star(dM + \log(|\Phi|/\delta))}{K}} + \frac{AdH^2(dM + \log(|\Phi|/\delta))}{K} \right).$$

*Proof.* As described above, we have $\mathrm{DE}_1(1/K) \leq \mathcal{O}(d \log(dK))$ and $\beta = \log(HK/\delta) + dM + \log|\Phi|$. Since DistBC is satisfied by $\mathcal{F}^{\mathrm{lin}}$, plugging into Theorem 5.5 gives the result. □

This is the first small-loss bound for low-rank MDPs, and for online RL with non-linear function approximation in general. Again when $V^\star \leq \widetilde{\mathcal{O}}(1/K)$, O-DISCO has a fast $\widetilde{\mathcal{O}}(1/K)$ convergence rate which improves over all prior results that converge at a slow $\widetilde{\Omega}(1/\sqrt{K})$ rate [Uehara et al., 2021].

## 5.4 Proof Sketch of Theorem 5.5

By DistBC (Assumption 5.1), we can deduce two facts about the construction of $\mathcal{F}_k$: (i) $Z^\star \in \mathcal{F}_k$, and (ii) elements of $\mathcal{F}_k$ almost satisfy the distributional Bellman equation, *i.e.*, for all $h \in [H]$, we have $\sum_{i=1}^k \mathbb{E}_{\pi^i}[\delta_{h,k}(x_h, a_h)] \leq \mathcal{O}(\beta)$ where $\delta_{h,k}(x_h, a_h) = D_\triangle(f_h^{(k)}(x_h, a_h) \ \| \ \mathcal{T}_h^{\star,D} f_{h+1}^{(k)}(x_h, a_h))$. Next, we derive a corollary of Eq. ($\triangle_1$):

$$|\bar f - \bar g| \leq \sqrt{4\bar g + D_\triangle(f \| g)} \cdot \sqrt{D_\triangle(f \| g)}. \tag{$\triangle_2$}$$

To see why this is true, apply AM-GM to Eq. ($\triangle_1$) to get $2(\bar f - \bar g) \leq \bar f + \bar g + D_\triangle(f \| g)$, which simplifies to $\bar f \leq 3\bar g + D_\triangle(f \| g)$. Plugging this back into Eq. ($\triangle_1$) yields Eq. ($\triangle_2$). Then, by iterating Eq. ($\triangle_2$) and AM-GM, we derive a self-bounding lemma: for any $f, \pi, h$, we have $\bar f_h(x_h, a_h) \lesssim Q_h^\pi(x_h, a_h) + H \sum_{t=h}^H \mathbb{E}_{\pi, x_h, a_h}[D_\triangle(f_t(x_t, a_t) \| \mathcal{T}_h^{\pi,D} f_{t+1}(x_t, a_t))]$ (Lemma H.3).

Since $\mathcal{T}_h^{\pi^k} \bar{f}_{h+1}^{(k)}(x,a) = \overline{\mathcal{T}_h^{\pi^k,D} f_{h+1}^{(k)}(x,a)}$ and $\mathcal{T}_h^{\pi^k,D} f_{h+1}^{(k)} = \mathcal{T}_h^{\star,D} f_{h+1}^{(k)}$, we have

$$V^{\pi^k} - V^\star \leq V^{\pi^k} - \bar{f}_1^{(k)}(x_1, \pi_1^k(x_1)) \qquad \text{(optimism from fact (i))}$$

$$= \sum_{h=1}^H \mathbb{E}_{\pi^k} \left[ \mathcal{T}_h^{\pi^k} \bar{f}_{h+1}^{(k)}(x_h, a_h) - \bar{f}_h^{(k)}(x_h, a_h) \right] \qquad \text{(performance difference)}$$

$$\leq 2 \sum_{h=1}^H \sqrt{\mathbb{E}_{\pi^k}[\bar{f}_h^{(k)}(x_h,a_h) + \delta_{h,k}(x_h,a_h)]} \sqrt{\mathbb{E}_{\pi^k}[\delta_{h,k}(x_h,a_h)]} \qquad \text{(Eq. ($\triangle_2$))}$$

$$\lesssim \sqrt{V^{\pi^k} w + H \sum_{h=1}^H \mathbb{E}_{\pi^k}[\delta_{h,k}(x_h,a_h)]} \sqrt{H \mathbb{E}_{\pi^k}[\delta_{h,k}(x_h,a_h)]}. \qquad \text{(Lemma H.3)}$$

The implicit inequality $V^{\pi^k} - V^\star \lesssim \sqrt{V^\star + H \sum_{h=1}^H \mathbb{E}_{\pi^k}[\delta_{h,k}(x_h,a_h)]} \sqrt{H \mathbb{E}_{\pi^k}[\delta_{h,k}(x_h,a_h)]}$ can then be obtained by AM-GM and rearranging. The final step is to sum over $k$ and bound $\sum_{k=1}^K \mathbb{E}_{\pi^k}[\delta_{h,k}(x_h,a_h)]$ via the eluder dimension's pigeonhole principle (Theorem 5.3 applied with fact (ii)). Please see Appendix H for the full proof.

## 6 Small-Loss Bounds for Offline Distributional RL

We now propose **Pe**ssimistic **Dis**tributional **C**onfidence set **O**ptimization (P-DISCO; Algorithm 3), which adapts the distributional confidence set technique from the previous section to the offline setting by leveraging pessimism instead of optimism. Notably, P-DISCO is a simple two-step algorithm that achieves the first small-loss PAC bounds in offline RL. First, construct a distributional confidence set for each policy $\pi$ based on a similar log-likelihood thresholding procedure as in O-DISCO, where the difference is we now use data sampled from $\mathcal{T}_h^{\pi,D} f_{h+1}$ instead of $\mathcal{T}_h^{\star,D} f_{h+1}$. Next, output the policy with the most pessimistic mean amongst all the confidence sets.

---

**Algorithm 3** **Pe**ssimistic **Dis**tributional **C**onfidence set **O**ptimization (P-DISCO)

---
1: **Input:** datasets $\mathcal{D}_1, \ldots, \mathcal{D}_H$, distribution function class $\mathcal{F}$, threshold $\beta$, policy class $\Pi$.
2: For all $(h, f, \pi) \in [H] \times \mathcal{F} \times \Pi$, sample $y_{h,i}^{f,\pi} \sim f_{h+1}(x'_{h,i}, \pi_{h+1}(x'_{h,i}))$, where $(x_{h,i}, a_{h,i}, c_{h,i}, x'_{h,i})$ is the $i$-th datapoint of $\mathcal{D}_h$. Then, set $z_{h,i}^{f,\pi} = c_{h,i} + y_{h,i}^{f,\pi}$ and define the confidence set,

$$\mathcal{F}_\pi = \left\{ f \in \mathcal{F} : \sum_{i=1}^N \log f_h(z_{h,i}^{f,\pi} \mid x_{h,i}, a_{h,i}) \geq \max_{g \in \mathcal{F}_h} \sum_{i=1}^N \log g(z_{h,i}^{f,\pi} \mid x_{h,i}, a_{h,i}) - 7\beta, \forall h \in [H] \right\}.$$

3: For each $\pi \in \Pi$, define the pessimistic estimate $f^\pi = \arg\max_{f \in \mathcal{F}_\pi} \mathbb{E}_{a \sim \pi(x_1)}[\bar{f}_1(x_1, a)]$.
4: **Output:** $\widehat{\pi} = \arg\max_{\pi \in \Pi} \mathbb{E}_{a \sim \pi(x_1)}[\bar{f}_1^\pi(x_1, \pi)]$.

---

In offline RL, many works made strong all-policy coverage assumptions [Antos et al., 2008, Chen and Jiang, 2019]. Recent advancements [Kidambi et al., 2020, Xie et al., 2021, Uehara and Sun, 2022, Rashidinejad et al., 2021, Jin et al., 2021b] have pursued *best effort* guarantees that aim to compete with any covered policy $\widetilde{\pi}$, with sub-optimality of the learned $\widehat{\pi}$ degrading gracefully as coverage worsens. The coverage is measured by the single-policy concentrability $C^{\widetilde{\pi}} = \max_h \left\| \mathrm{d}d_h^{\widetilde{\pi}}/\mathrm{d}\nu_h \right\|_\infty$. We adopt this framework and obtain the first small-loss PAC bound in offline RL.

**Theorem 6.1** (Small-Loss PAC bound for P-DISCO). *Assume Assumption 5.1. For any $\delta \in (0,1)$, w.p. at least $1 - \delta$, running P-DISCO with $\beta = \log(H|\Pi||\mathcal{F}|/\delta)$ learns a policy $\widehat{\pi}$ that enjoys the following PAC bound with respect to any comparator policy $\widetilde{\pi} \in \Pi$:*

$$V^{\widehat{\pi}} - V^{\widetilde{\pi}} \leq 9H \sqrt{\frac{C^{\widetilde{\pi}} V^{\widetilde{\pi}} \beta}{N}} + \frac{30H^2 C^{\widetilde{\pi}} \beta}{N}.$$

To the best of our knowledge, this is the first small-loss bound for offline RL, which we highlight illustrates a novel robustness property against bad coverage. Namely, the dominant term not only scales with the coverage coefficient $C^{\widetilde{\pi}}$ but also the comparator policy's value $V^{\widetilde{\pi}}$. In particular, P-DISCO can strongly compete with a comparator policy $\widetilde{\pi}$ if *one of the following* is true: (i) $\nu$ has good coverage over $\widetilde{\pi}$, so the $\mathcal{O}(1/\sqrt{N})$ term is manageable; *or* (ii) $\widetilde{\pi}$ has small-loss, in which case we may even obtain a fast $\mathcal{O}(1/N)$ rate. Thus, P-DISCO has *two* chances at strongly competing with $\widetilde{\pi}$, while conventional offline RL methods solely rely on (i) to be true.

# 7 Distributional CB Experiments

We now compare DISTCB with SquareCB [Foster and Rakhlin, 2020] and the state-of-the-art CB method FastCB [Foster and Krishnamurthy, 2021], which respectively minimize the squared loss and log loss for estimating the conditional mean. The key question we investigate here is whether learning the conditional mean via distribution learning with MLE will demonstrate empirical benefit over the non-distributional approaches. We consider three challenging tasks that are all derived from real-world datasets and we briefly describe the construction below.

| Algorithm: | SquareCB | FastCB | DistCB (Ours) |
|---|---|---|---|
| King County Housing [Vanschoren et al., 2013] | | | |
| All episodes | .756 (.0007) | .734 (.0007) | **.726** (.0003) |
| Last 100 ep. | .725 (.0012) | .719 (.0013) | **.708** (.0019) |
| Prudential Life Insurance [Montoya et al., 2015] | | | |
| All episodes | .456 (.0082) | .491 (.0029) | **.411** (.0038) |
| Last 100 ep. | .481 (.0185) | .474 (.0111) | **.388** (.0086) |
| CIFAR-100 [Krizhevsky, 2009] | | | |
| All episodes | .872 (.0010) | .856 (.0016) | **.838** (.0021) |
| Last 100 ep. | .828 (.0024) | .793 (.0031) | **.775** (.0027) |

Table 1: Avg cost over all episodes and last 100 episodes (lower is better). We report 'mean (sem)' over 10 seeds.

**King County Housing** This dataset consists of home features and prices, which we normalize to be in $[0, 1]$. The action space is 100 evenly spaced prices between $0.01$ and $1.0$. If the learner overpredicts the true price, the cost is $1.0$. Else, the cost is $1.0$ minus predicted price.

**Prudential Life Insurance** This dataset contains customer features and an integer risk level in $[8]$, which is our action space. If the model overpredicts the risk level, the cost is $1.0$. Otherwise, the cost is $.1 \times (y - \hat{y})$ where $y$ is the actual risk level, and $\hat{y}$ is the predicted risk level.

**CIFAR-100** This popular image dataset contains 100 classes, which correspond to our actions, and each class is in one of 20 superclasses. We assign cost as follows: $0.0$ for predicting the correct class, $0.5$ for the wrong class but correct superclass, and $1.0$ for a fully incorrect prediction.

**Results** Across tasks, DISTCB achieves lower average cost over all episodes (*i.e.*, normalized regret) and over the last 100 episodes (*i.e.*, most updated policies' performance) compared to SquareCB. This indicates the empirical benefit of the distributional approach over the conventional approach based on least square regression, matching the theoretical benefit demonstrated here. Perhaps surprisingly, DISTCB also consistently outperforms FastCB. Both methods obtain first-order bounds with the same dependencies on $A$ and $C^\star$, which suggests that DISTCB's empirical improvement over FastCB cannot be fully explained by existing theory. The only difference between DISTCB and FastCB is that the former integrates online MLE while the latter directly estimates the mean by minimizing the log loss (binary cross-entropy). An even more fine-grained understanding of the benefits of distribution learning may therefore be helpful in explaining this improvement. Appendix K contains all experiment details. Reproducible code is available at https://github.com/kevinzhou497/distcb.

# 8 Conclusion

We showed that distributional RL leads to small-loss bounds in both online and offline RL, and we also proposed a distributional CB algorithm that outperforms the state-of-the-art FastCB. A fruitful direction would be to investigate connections of natural policy gradient with our MLE distributional-fitting scheme to inspire a practical offline RL algorithm with small loss guarantees, *à la* Cheng et al. [2022]. Finally, it would be interesting to investigate other loss functions that yield small-loss or even faster bounds.

**Acknowledgements** This material is based upon work supported by the National Science Foundation under Grant Nos. IIS-1846210 and IIS-2154711.

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

# Appendices

## A   Notations

Table 2: List of Notations

| | |
|---|---|
| $\mathcal{S}, \mathcal{A}, A$ | State and action spaces, and $A = |\mathcal{A}|$. |
| $\Delta(S)$ | The set of distributions supported by $S$. |
| $\bar{d}$ | The expectation of any real-valued distribution $d$, *i.e.*, $\bar{d} = \mathbb{E}_{y \sim d}[y]$. |
| $[N]$ | $\{1, 2, \dots, N\}$ for any natural number $N$. |
| $Z_h^\pi(x, a)$ | Distribution of $\sum_{t=h}^{H} c_t$ given $x_h = x, a_h = a$ rolling in from $\pi$. |
| $Q_h^\pi(x, a), V_h^\pi(x)$ | $Q_h^\pi(x, a) = \bar{Z}_h^\pi(x, a)$ and $V_h^\pi = \mathbb{E}_{a \sim \pi(x)}[Q_h^\pi(x, a)]$. |
| $\pi^\star$ | Optimal policy, *i.e.*, $\pi^\star = \arg\min_\pi V_1^\pi(x_1)$. |
| | Without loss of optimality, we take $\pi^\star : \mathcal{X} \to \mathcal{A}$ to be Markov & deterministic. |
| $Z_h^\star, Q_h^\star, V_h^\star$ | $Z_h^\pi, Q_h^\pi, V_h^\pi$ with $\pi = \pi^\star$, the optimal policy. |
| $\mathcal{T}_h^\pi, \mathcal{T}_h^\star$ | The Bellman operators that act on functions. |
| $\mathcal{T}_h^{\pi,D}, \mathcal{T}_h^{\star,D}$ | The distributional Bellman operators that act on conditional distributions. |
| $V^\pi, Z^\pi, V^\star, Z^\star$ | $V^\pi = V_1^\pi(x_1), Z^\pi = Z_1^\pi(x_1)$. $V^\star, Z^\star$ are defined similarly with $\pi^\star$. |
| $d_h^\pi(x, a)$ | The probability of $\pi$ visiting $(x, a)$ at time $h$. |
| $C^{\tilde{\pi}}$ | Coverage coefficient $\max_h \left\| \mathrm{d} d_h^{\tilde{\pi}}/\mathrm{d}\nu_h \right\|_\infty$. |
| $D_\triangle(f \parallel g)$ | Triangular discrimination between $f, g$. |
| $H(f \parallel g)$ | Hellinger distance between $f, g$. |
| $D_{KL}(f \parallel g)$ | KL divergence between $f, g$. |

### A.1   Statistical Distances

Let $f, g$ be distributions over $\mathcal{Y}$. Then,

$$D_\triangle(f \parallel g) = \sum_y \frac{(f(y) - g(y))^2}{f(y) + g(y)},$$

$$H(f \parallel g) = \sqrt{\frac{1}{2} \sum_y \left( \sqrt{f(y)} - \sqrt{g(y)} \right)^2},$$

$$D_{KL}(f \parallel g) = \sum_y f(y) \log(f(y)/g(y)),$$

$$D_{TV}(f \parallel g) = \frac{1}{2} \sum_y |f(y) - g(y)|.$$

The following standard inequalities will be helpful:

$$H^2 \leq D_{TV} \leq \sqrt{2} H,$$
$$2H^2 \leq D_\triangle \leq 4H^2, \qquad \text{(Lemma A.1)}$$
$$H \leq \sqrt{D_{KL}}.$$

**Lemma A.1.** *For any distributions $f, g$, we have $2H^2(f \parallel g) \leq D_\triangle(f \parallel g) \leq 4H^2(f \parallel g)$.*

*Proof.* Recall that

$$D_\triangle(f \parallel g) = \int_y \left( \frac{f(y) - g(y)}{\sqrt{f(y) + g(y)}} \right)^2.$$

Applying $\frac{1}{\sqrt{f(y)} + \sqrt{g(y)}} \leq \frac{1}{\sqrt{f(y) + g(y)}} \leq \frac{\sqrt{2}}{\sqrt{f(y)} + \sqrt{g(y)}}$ concludes the proof. $\qquad\square$

# B Modified Algorithms with UAE and for Small Returns Bounds

In this section, we present the O-DISCO algorithm with Uniform Action Exploration (UAE). We also present versions of O-DISCO and P-DISCO for the reward-maximizing setting (instead of the cost-minimizing setting studied throughout the paper); if SMALLRETURN is turned on, we can derive small-return bounds in Appendix J.

---

**Algorithm 4** O-DISCO (with UAE and small return)

---

1: **Input:** number of episodes $K$, distribution function class $\mathcal{F}$, threshold $\beta$, flag UAE, flag SMALLRETURN.
2: Initialize $\mathcal{D}_{h,0} \leftarrow \emptyset$ for all $h \in [H]$, and set $\mathcal{F}_0 = \mathcal{F}$.
3: Set $\mathrm{op} = \max$ if SMALLRETURN else $\mathrm{op} = \min$.
4: **for** episode $k = 1, 2, \ldots, K$ **do**
5:      Set $f^{(k)} = \arg \mathrm{op}_{f \in \mathcal{F}_{k-1}} \mathrm{op}_a \bar{f}_1(x_1, a)$.
6:      Set $\pi_h^k(x) = \arg \mathrm{op}_a \bar{f}_h^{(k)}(x, a)$.
7:      **if** UAE **then**
8:          For each $h \in [H]$, collect $x_{h,k} \sim d_h^{\pi^k}, a_{h,k} \sim \mathrm{unif}(\mathcal{A}), c_{h,k} \sim C_h(x_{h,k}, a_{h,k}), x'_{h,k} \sim$
         $P_h(x_{h,k}, a_{h,k})$, and augment the dataset $\mathcal{D}_{h,k} = \mathcal{D}_{h,k-1} \cup \left\{(x_{h,k}, a_{h,k}, c_{h,k}, x'_{h,k})\right\}$.
9:      **else**
10:          Roll out $\pi^k$ and obtain a trajectory $x_{1,k}, a_{1,k}, c_{1,k}, \ldots, x_{H,k}, a_{H,k}, c_{H,k}$.
         For each $h \in [H]$, augment the dataset $\mathcal{D}_{h,k} = \mathcal{D}_{h,k-1} \cup \left\{(x_{h,k}, a_{h,k}, c_{h,k}, x_{h+1,k})\right\}$.
11:      **end if**
12:      For all $(h, f) \in [H] \times \mathcal{F}$, sample $y_{h,i}^f \sim f_{h+1}(x'_{h,i}, a')$ and $a' = \arg \mathrm{op}_a \bar{f}_{h+1}(x'_{h,i}, a)$,
     where $(x_{h,i}, a_{h,i}, c_{h,i}, x'_{h,i})$ is the $i$-th datapoint of $\mathcal{D}_{h,k}$. Also, set $z_{h,i}^f = c_{h,i} + y_{h,i}^f$ and define the confidence set,

$$\mathcal{F}_k = \left\{ f \in \mathcal{F} : \sum_{i=1}^k \log f_h(z_{h,i}^f \mid x_{h,i}, a_{h,i}) \geq \max_{\widetilde{f} \in \mathcal{F}} \sum_{i=1}^k \log \widetilde{f}_h(z_{h,i}^f \mid x_{h,i}, a_{h,i}) - 7\beta, \forall h \in [H] \right\}.$$

13: **end for**
14: **Output:** $\bar{\pi} = \mathrm{unif}(\pi^{1:K})$.

---

---

**Algorithm 5** P-DISCO (with small return)

---

1: **Input:** datasets $\mathcal{D}_1, \ldots, \mathcal{D}_H$, distribution function class $\mathcal{F}$, threshold $\beta$, policy class $\Pi$, flag SMALLRETURN.
2: For all $(h, f, \pi) \in [H] \times \mathcal{F} \times \Pi$, sample $y_{h,i}^{f,\pi} \sim f_{h+1}(x'_{h,i}, \pi_{h+1}(x'_{h,i}))$, where $(x_{h,i}, a_{h,i}, c_{h,i}, x'_{h,i})$ is the $i$-th datapoint of $\mathcal{D}_h$. Then, set $z_{h,i}^{f,\pi} = c_{h,i} + y_{h,i}^{f,\pi}$ and define the confidence set,

$$\mathcal{F}_\pi = \left\{ f \in \mathcal{F} : \sum_{i=1}^N \log f_h(z_{h,i}^{f,\pi} \mid x_{h,i}, a_{h,i}) \geq \max_{\widetilde{f} \in \mathcal{F}} \sum_{i=1}^N \log \widetilde{f}_h(z_{h,i}^{f,\pi} \mid x_{h,i}, a_{h,i}) - 7\beta, \forall h \in [H] \right\}.$$

3: Set $\mathrm{op} = \max$ if SMALLRETURN else $\mathrm{op} = \min$.
4: For each $\pi \in \Pi$, define the pessimistic estimate $f^\pi = \arg \mathrm{op}_{f \in \mathcal{F}_\pi} \mathbb{E}_{a \sim \pi(x_1)}\left[\bar{f}_1(x_1, a)\right]$.
5: **Output:** $\widehat{\pi} = \arg \mathrm{op}_{\pi \in \Pi} \mathbb{E}_{a \sim \pi(x_1)}\left[\bar{f}_1^\pi(x_1, \pi)\right]$.

---

## C Proofs for DISTCB

**Lemma C.1** (Azuma). *Let* $\{X_i\}_{i \in [N]}$ *be a sequence of random variables supported on* $[0, 1]$, *adapted to filtration* $\{\mathcal{F}_i\}_{i \in [N]}$. *For any* $\delta \in (0, 1)$, *we have w.p. at least* $1 - \delta$,

$$\sum_{t=1}^{N} \mathbb{E}[X_t \mid \mathcal{F}_{t-1}] \leq \sum_{t=1}^{N} X_t + \sqrt{N \log(2/\delta)}, \qquad \text{(Standard Azuma)}$$

$$\sum_{t=1}^{N} \mathbb{E}[X_t \mid \mathcal{F}_{t-1}] \leq 2 \sum_{t=1}^{N} X_t + 2 \log(1/\delta). \qquad \text{(Multiplicative Azuma)}$$

*Proof.* For standard Azuma, see Zhang [2023, Theorem 13.4]. For multiplicative Azuma, apply [Zhang, 2023, Theorem 13.5] with $\lambda = 1$. The claim follows, since $\frac{1}{1-\exp(-\lambda)} \leq 2$. $\qquad \square$

**Theorem 4.1.** *For any* $\delta \in (0, 1)$, *w.p. at least* $1 - \delta$, *running* DISTCB *with* $\gamma = 10A \vee \sqrt{\frac{40A(C^\star + \log(1/\delta))}{112(\mathrm{Regret}_{\log}(K) + \log(1/\delta))}}$ *has regret scaling with* $C^\star = \sum_{k=1}^{K} \min_{a \in \mathcal{A}} \bar{C}(x_k, a)$,

$$\mathrm{Regret}_{\mathrm{DISTCB}}(K) \leq 232 \sqrt{AC^\star \, \mathrm{Regret}_{\log}(K) \log(1/\delta)} + 2300A\big(\mathrm{Regret}_{\log}(K) + \log(1/\delta)\big).$$

*Proof of Theorem 4.1.* First, recall the per-step inequality of ReIGW Foster and Krishnamurthy [2021, Theorem 4], which states: for any $\widehat{f}$ and $\gamma \geq 2A$, if we set $p = \mathrm{ReIGW}_\gamma(\widehat{f}, \gamma)$, then, for all $f \in [0, 1]^A$, we have

$$\sum_a p(a)(f(a) - f(a^\star)) \leq \frac{5A}{\gamma} \sum_a p(a)f(a) + 7\gamma \sum_a p(a) \frac{(\widehat{f}(a) - f(a))^2}{\widehat{f}(a) + f(a)},$$

where $a^\star = \arg\min_a f(a)$. For any $k \in [K]$, applying this to $\widehat{f} = \bar{f}^{(k)}(s_k, \cdot)$, $p = p_k$ and $f = \bar{C}(s_k, \cdot)$, we have

$$\sum_{k=1}^{K} \mathbb{E}_{a_k}\big[\bar{C}(s_k, a_k) - \bar{C}(s_k, \pi^\star(s_k))\big] \leq \sum_{k=1}^{K} \mathbb{E}_{a_k}\left[\frac{5A}{\gamma}\bar{C}(s_k, a_k) + 7\gamma \frac{\big(\bar{f}^{(k)}(s_k, a_k) - \bar{C}(s_k, a_k)\big)^2}{\bar{f}^{(k)}(s_k, a_k) + \bar{C}(s_k, a_k)}\right]$$

$$\leq \sum_{k=1}^{K} \mathbb{E}_{a_k}\left[\frac{5A}{\gamma}\bar{C}(s_k, a_k) + 7\gamma D_\triangle(f^{(k)}(s_k, a_k) \parallel C(s_k, a_k))\right]$$

$$\text{(Eq. } (\triangle_1))$$

Since $D_\triangle \leq 4H^2$, we have

$$\sum_{k=1}^{K} \mathbb{E}_{a_k}\Big[D_\triangle(f^{(k)}(s_k, a_k) \parallel C(s_k, a_k))\Big]$$

$$\leq 4 \sum_{k=1}^{K} \mathbb{E}_{a_k}\Big[H^2\Big(C(s_k, a_k) \parallel f^{(k)}(s_k, a_k)\Big)\Big]$$

$$\leq 8 \sum_{k=1}^{K} H^2\Big(C(s_k, a_k) \parallel f^{(k)}(s_k, a_k)\Big) + 8\log(1/\delta) \quad \text{(Multiplicative Azuma, since } H^2 \in [0, 1])$$

$$\leq 8\,\mathrm{Regret}_{\log}(K) + 10\log(1/\delta). \qquad \text{(Foster et al. [2021, Lemma A.14])}$$

Hence, we have

$$\sum_{k=1}^{K} \mathbb{E}_{a_k}\big[\bar{C}(s_k, a_k) - \bar{C}(s_k, \pi^\star(s_k))\big] \leq \frac{5A}{\gamma} \sum_{k=1}^{K} \mathbb{E}_{a_k}\big[\bar{C}(s_k, a_k)\big] + 70\gamma\big(\mathrm{Regret}_{\log}(K) + \log(1/\delta)\big).$$

Finally, recalling that $1/(1 - \varepsilon) \leq 1 + 2\varepsilon$ when $\varepsilon \leq \frac{1}{2}$, and the fact that $\frac{5A}{\gamma} \leq \frac{1}{2}$, we have

$$\sum_{k=1}^{K} \mathbb{E}_{a_k}\big[\bar{C}(s_k, a_k) - \bar{C}(s_k, \pi^\star(s_k))\big] \leq \frac{10A}{\gamma} \sum_{k=1}^{K} \mathbb{E}_{a_k}\big[\bar{C}(s_k, \pi^\star(s_k))\big] + 140\gamma\big(\mathrm{Regret}_{\log}(K) + \log(1/\delta)\big).$$

By Azuma's inequality, we have

$$\sum_{k=1}^{K} \bar{C}(s_k, a_k) - \bar{C}(s_k, \pi^\star(s_k))$$

$$\leq 2 \sum_{k=1}^{K} \mathbb{E}_{a_k}\big[\bar{C}(s_k, a_k) - \bar{C}(s_k, \pi^\star(s_k))\big] + 2\log(1/\delta)$$

$$\leq \frac{20A}{\gamma} \sum_{k=1}^{K} \mathbb{E}_{a_k}\big[\bar{C}(s_k, \pi^\star(s_k))\big] + 140\gamma\big(\mathrm{Regret}_{\log}(K) + \log(1/\delta)\big) + 2\log(1/\delta)$$

$$\leq \frac{40A}{\gamma}\big(C^\star + \log(1/\delta)\big) + 140\gamma\big(\mathrm{Regret}_{\log}(K) + \log(1/\delta)\big) + 2\log(1/\delta).$$

(Multiplicative Azuma)

Now set $\gamma = \sqrt{\frac{40A(C^\star + \log(1/\delta))}{140\big(\mathrm{Regret}_{\log}(K) + \log(1/\delta)\big)}} \vee 10A$.

Case 1 is when $\sqrt{\frac{40A(C^\star + \log(1/\delta))}{140\big(\mathrm{Regret}_{\log}(K) + \log(1/\delta)\big)}} \leq 10A$, *i.e.*, $(C^\star + \log(1/\delta)) \leq 280A\big(\mathrm{Regret}_{\log}(K) + \log(1/\delta)\big)$, we have the above is at most

$$4(C^\star + \log(1/\delta)) + 1120A\big(\mathrm{Regret}_{\log}(K) + \log(1/\delta)\big) + 2\log(1/\delta)$$
$$\leq 2240A\big(\mathrm{Regret}_{\log}(K) + \log(1/\delta)\big) + 2\log(1/\delta).$$

Case 2 is when the left term dominates, then the bound is,

$$2\sqrt{4480A(C^\star + \log(1/\delta))\big(\mathrm{Regret}_{\log}(K) + \log(1/\delta)\big)} + 2\log(1/\delta)$$

$$\leq 2\sqrt{13440AC^\star \, \mathrm{Regret}_{\log}(K)\log(1/\delta) + 4480A\log^2(1/\delta)} + 2\log(1/\delta)$$

$$\leq 232\sqrt{AC^\star \, \mathrm{Regret}_{\log}(K)\log(1/\delta)} + 134\sqrt{A}\log(1/\delta) + 2\log(1/\delta).$$

Putting these two cases together, we have the result. $\qquad\square$

# D   Distributional Bellman Completeness in low-rank MDPs

The goal of this section is to show that, under mild conditions in low-rank MDPs, there always exists a function class with bounded bracketing number that satisfies the distributional BC condition. First, let us recall the low-rank MDP In this section, we show that linear MDPs automatically satisfy the distributional Bellman completeness assumption.

**Definition 5.6** (Low-rank MDP). A transition model $P_h : \mathcal{X} \times \mathcal{A} \to \Delta(\mathcal{X})$ has rank $d$ if there exist unknown features $\phi_h^\star : \mathcal{X} \times \mathcal{A} \to \mathbb{R}^d, \mu_h^\star : \mathcal{X} \to \mathbb{R}^d$ such that $P_h(x' \mid x, a) = \phi_h^\star(x, a)^\top \mu_h^\star(x')$ for all $x, a, x'$. Also, assume $\max_{x,a} \|\phi_h^\star(x, a)\|_2 \leq 1$ and $\|\int g \mathrm{d}\mu_h^\star\|_2 \leq \|g\|_\infty \sqrt{d}$ for all functions $g : \mathcal{X} \to \mathbb{R}$. The MDP is called low-rank if $P_h$ is low-rank for all $h \in [H]$.

Suppose that we have a function class $\Phi$ such that $\phi_h^\star \in \Phi$ for all $h$, *i.e.*, $\Phi$ is a realizable function class. For example, in linear MDPs, this is automatically satisfied since we know $\phi^\star$ *a priori*, so $\Phi$ is the singleton with $\phi^\star$. Having a realizable $\Phi$ class is standard for solving low-rank MDPs [Uehara et al., 2021, Agarwal et al., 2023].

In what follows, let $\alpha = \max_{h,\pi,z,x,a} Z_h^\pi(z \mid x, a)$ denote the maximum density/mass value of the loss-to-go distributions. Note that $\alpha \geq 1$ always since the mass at $H + 1$ is deterministically placed at zero. If we further know that $Z_h^\pi$ is discretely distributed, then $\alpha = 1$. If $Z_h^\pi$ is continuously distributed, we assume it is bounded.

We consider the function class in Eq. (2), which we reproduce here:

$$\mathcal{F}_h^{\mathrm{lin}} = \left\{ f(z \mid x, a) = \langle \phi(x, a), w(z) \rangle \quad : \quad \phi \in \Phi, w : [0, 1] \to \mathbb{R}^d, \right. \tag{3}$$

$$\left. \text{s.t. } \max_z \|w(z)\|_2 \leq \alpha \sqrt{d} \ \text{ and } \ \max_{x,a,z} \langle \phi(x, a), w(z) \rangle \leq \alpha \right\}.$$

The next lemma (Lemma D.1) shows that this function class satisfies distributional BC.

**Lemma D.1.** $\mathcal{F}^{lin}$ *satisfies distributional BC (Assumption 5.1).*

*Proof.* We denote $\|f\|_\infty = \max_{z,x,a} f(z \mid x, a)$. For any $f_{h+1} \in \mathcal{F}_{h+1}^{\mathrm{lin}}$, we have $\|f_{h+1}\|_\infty \leq \alpha$ by the construction of $\mathcal{F}_{h+1}^{\mathrm{lin}}$. Then, let $\mathcal{T}^D$ be either the distributional Bellman operator or distributional optimality operator, the following equalities hold for the appropriate $a'(x')$ based on $\mathcal{T}^D$,

$$\mathcal{T}^D f_{h+1}(z \mid x, a) = \int_{\mathcal{X}} \Pr_h(x' \mid x, a) \int_{\mathbb{R}} \Pr_h(c \mid x, a) f_{h+1}(z - c \mid x', a'(x')) \, \mathrm{d}x' \, \mathrm{d}c$$

$$= \left\langle \phi_h^\star(x, a), \ \underbrace{\int_{\mathcal{X}} \mu_h(x') \int_{\mathbb{R}} \Pr_h(c \mid x, a) f_{h+1}(z - c \mid x', a'(x')) \, \mathrm{d}c \, \mathrm{d}x'}_{:=w_h(z)} \right\rangle$$

Since $\int_{\mathbb{R}} \Pr_h(c \mid x, a) f_{h+1}(z - c \mid x', a'(x')) \, \mathrm{d}c \leq \|f_{h+1}\|_\infty$, we know that

$$\|w_h(z)\|_2 \leq \|f_{h+1}\|_\infty \sqrt{d} \leq \alpha \sqrt{d}.$$

We further note that

$$\max_{x,a,z} \langle \phi_h^\star(x, a), w_h(z) \rangle = \max_{x,a,z} \mathcal{T}^D f_{h+1}(z \mid x, a) \leq \|f_{h+1}\|_\infty \leq \alpha.$$

Also note that $\phi_h^\star \in \Phi$ by realizability. Therefore, $\mathcal{T}^D f_{h+1} \in \mathcal{F}_h^{\mathrm{lin}}$, which is the distributional BC condition. $\square$

## D.1   Bounding the bracketing number via discretized rewards

We now bound the bracketing number of $\mathcal{F}_h^{\mathrm{lin}}$ under a *discretization assumption that costs and costs-to-gos can only take $M$ many discrete values on an evenly spaced grid*. This can be interpreted as discretizing the reward space, and it can be shown that this discretization error is small for regret or PAC bounds [Wang et al., 2023, Section 6]. Structural assumptions are necessary to bound the

complexity of $\mathcal{F}_h^{\text{lin}}$ and such discretization assumptions are common in practice, *e.g.*, C51 [Bellemare et al., 2017] and Rainbow [Hessel et al., 2018] both set $M = 51$ which works well in Atari games. After discretizing, we can consider $w$ as a mapping from $[M]$, the discrete set on $M$ elements, rather than from the interval $[0, 1]$. Note also that since $Z_h^\pi$ are discrete, we have $\alpha = 1$.

Now, let $\varepsilon > 0$ be arbitrary and fixed. Recall that the $\ell_\infty$ bracketing number is equivalent (up to universal constants) to the $\ell_\infty$ covering number, so we will work with the latter. Let $B(r)$ denote the $d$-dimensional ball of radius $r$ (in $\ell_2$). Recall that the $\varepsilon$-covering number (in $\ell_2$) of functions $[M] \mapsto B(r)$ scales as $\mathcal{O}((r/\varepsilon)^{dM})$. Let $\mathcal{W}_\varepsilon$ be such the smallest cover. We can build a $\ell_\infty$ cover of $\mathcal{F}_h^{\text{lin}}$ as follows: $\mathcal{C}_\varepsilon = \{(x, a, z) \mapsto \langle \phi(x, a), w(z) \rangle, w \in \mathcal{W}_\varepsilon, \phi \in \Phi\}$.

To check this is a $\varepsilon$ cover, consider any $f \in \mathcal{F}_h^{\text{lin}}$. $f$ corresponds to some $\phi$ and $w$. Let $w'$ be the neighbor of $w$ in $\mathcal{W}_\varepsilon$ and let $f'(x, a, z) = \langle \phi(x, a), w'(z) \rangle$ so indeed $f' \in \mathcal{C}_\varepsilon$. Then, for any $x, a, z$, we have $|\langle \phi(x, a), w(z) - w'(z) \rangle| \leq \|\phi(x, a)\|_2 \|w(z) - w'(z)\|_2 \leq \varepsilon$. Hence, $\mathcal{C}_\varepsilon$ is an $\ell_\infty$ cover of size $\mathcal{O}((\sqrt{d}/\varepsilon)^{dM} \cdot |\Phi|)$, and so we have shown that $\log N_{[]}(\varepsilon, \mathcal{F}_h^{\text{lin}}, \|\cdot\|_\infty) \leq \mathcal{O}(dM \log(d/\varepsilon) + \log |\Phi|)$.

**Linear MDPs:** Recall that in linear MDPs, we know the true $\phi^\star$ and so $|\Phi| = 1$. Thus, the bracketing number is simply $\mathcal{O}(dM \log(d/\varepsilon))$ in linear MDPs.

**Summary and comparison with regular BC:** In summary, under the assumption that rewards are discretized, we know that low-rank MDPs automatically have distributional function classes that satisfy distributional BC and have bounded bracketing numbers. Furthermore, recall that [Wu et al., 2023] showed that Linear Quadratic Regulators (LQRs), with deterministic transitions, also have function classes that satisfy distributional BC and have bounded bracketing numbers. Thus, distributional BC holds for the most interesting cases covered by the standard Bellman completeness, *e.g.*, linear MDPs, low-rank MDPs and LQRs. Since learning conditional distributions is statistically harder than learning the conditional mean, we need to pay the price in assuming reward/transitions satisfy regularity assumptions to bound the bracketing number appropriately.

# E    Generalization Bounds for Maximum Likelihood Estimation

This section reviews generalization bounds for the maximum likelihood estimator (MLE). We adopt the same sequential condition probability estimation setup as in Agarwal et al. [2020, Appendix E], which we now recall for completeness. Let $\mathcal{X}$ be the context/feature space and $\mathcal{Y}$ be the label space, and we are given a dataset $D = \{(x_i, y_i)\}_{i \in [n]}$ from a martingale process: for $i = 1, 2, ..., n$, sample $x_i \sim \mathcal{D}_i(x_{1:i-1}, y_{1:i-1})$ and $y_i \sim p(\cdot \mid x_i)$. Let $f^\star(x, y) = p(y \mid x)$ and we are given a realizable, i.e., $f^\star \in \mathcal{F}$, function class $\mathcal{F} : \mathcal{X} \times \mathcal{Y} \to \Delta(\mathbb{R})$ of distributions. The MLE is an estimate for $f^\star$ that maximizes the log-likelihood objective over our dataset:

$$\widehat{f}_{\mathrm{MLE}} = \arg\max_{f \in \mathcal{F}} \sum_{i=1}^{n} \log f(x_i, y_i).$$

For our guarantees to hold for general hypotheses classes $\mathcal{F}$, we use the bracketing number to quantify the statistical complexity of $\mathcal{F}$ [van de Geer, 2000].

**Definition E.1** (Bracketing Number). Let $\mathcal{G}$ be a set of functions mapping $\mathcal{X} \to \mathbb{R}$. Given two functions $l, u$ such that $l(x) \le u(x)$ for all $x \in \mathcal{X}$, the bracket $[l, u]$ is the set of functions $g \in \mathcal{G}$ such that $l(x) \le g(x) \le u(x)$ for all $x \in \mathcal{X}$. We call $[l, u]$ an $\varepsilon$-bracket if $\|u - l\| \le \varepsilon$. Then, the $\varepsilon$-bracketing number of $\mathcal{G}$ with respect to $\|\cdot\|$, denoted by $N_{[]}(\varepsilon, \mathcal{G}, \|\cdot\|)$ is the minimum number of $\varepsilon$-brackets needed to cover $\mathcal{G}$.

Since the triangular discrimination is equivalent to squared Hellinger up to universal constants, we now prove MLE generalization bounds in terms of squared Hellinger.

**Lemma E.2.** *Let $f_1 : \mathcal{X} \to \Delta(\mathcal{Y})$ and $f_2 : \mathcal{X} \times \mathcal{Y} \to \mathbb{R}_+$ satisfying $\sup_{x \in \mathcal{X}} \int_{\mathcal{Y}} f_2(x, y) \mathrm{d}y \le s$, then for any distribution $\mathcal{D} \in \Delta(\mathcal{X})$, we have*

$$\mathbb{E}_{x \sim \mathcal{D}}\big[H^2(f_1(x) \parallel f_2(x, \cdot))\big] \le (s - 1) - 2 \log \mathbb{E}_{x \sim \mathcal{D}, y \sim f_1(x)} \exp\left(-\frac{1}{2} \log(f_1(x, y)/f_2(x, y))\right).$$

*Proof.* This follows from the proof of Wu et al. [2023, Lemma C.1]. $\qquad\square$

**Lemma E.3.** *Fix $\delta \in (0, 1)$. Then w.p. at least $1 - \delta$, for any $f \in \mathcal{F}$, we have*

$$\sum_{i=1}^{n} \mathbb{E}_{x \sim \mathcal{D}_i}\big[H^2(f(x, \cdot) \parallel f^\star(x, \cdot))\big]$$

$$\le 6n\epsilon|\mathcal{Y}| + 2 \sum_{i=1}^{n} \log\big(f^\star(x_i, y_i)/f(x_i, y_i)\big) + 8 \log\big(N_{[]}(\epsilon, \mathcal{F}, \|\cdot\|_\infty)/\delta\big). \tag{4}$$

*Rearranging, we also have*

$$\sum_{i=1}^{n} \log\big(f(x_i, y_i)/f^\star(x_i, y_i)\big) \le 3n\epsilon|\mathcal{Y}| + 4 \log\big(N_{[]}(\epsilon, \mathcal{F}, \|\cdot\|_\infty)/\delta\big). \tag{5}$$

*Proof.* We take an $\epsilon$-bracketing of $\mathcal{F}$, $\{[l_i, u_i] : i = 1, 2, \dots\}$, and denote $\widetilde{\mathcal{F}} = \{u_i : i = 1, 2, \dots\}$. Applying Lemma 24 of Agarwal et al. [2020] to function class $\widetilde{\mathcal{F}}$ and using Chernoff method, w.p. at least $1 - \delta$, for all $\tilde{f} \in \widetilde{\mathcal{F}}$, we have

$$\underbrace{-\log \mathbb{E}_{D'} \exp(L(\tilde{f}(D), D'))}_{\text{(i)}} \le \underbrace{-L(\tilde{f}(D), D) + 2 \log\big(N_{[]}(\epsilon, \mathcal{F}, \|\cdot\|_\infty)/\delta\big)}_{\text{(ii)}}. \tag{6}$$

Now, fix any $f \in \mathcal{F}$ and pick $\tilde{f} \in \widetilde{\mathcal{F}}$ as the upper bracket, i.e., $f \le \tilde{f}$. Now set $L(f, D) = \sum_{i=1}^{n} -1/2 \log(f^\star(x_i, y_i)/f(x_i, y_i))$. Then the right hand side of (6) is

$$\text{(ii)} = \frac{1}{2} \sum_{i=1}^{n} \log(f^\star(x_i, y_i)/\tilde{f}(x_i, y_i)) + 2 \log\big(N_{[]}(\epsilon, \mathcal{F}, \|\cdot\|_\infty)/\delta\big)$$

$$\le \frac{1}{2} \sum_{i=1}^{n} \log(f^\star(x_i, y_i)/f(x_i, y_i)) + 2 \log\big(N_{[]}(\epsilon, \mathcal{F}, \|\cdot\|_\infty)/\delta\big).$$

On the other hand, since $H$ is a metric, we have

$$\sum_{i=1}^{n} \mathop{\mathbb{E}}_{x\sim\mathcal{D}_i} H^2\left(f(x,\cdot), f^\star(x,\cdot)\right) \leq \sum_{i=1}^{n} \mathop{\mathbb{E}}_{x\sim\mathcal{D}_i}\left(H\left(f(x,\cdot),\tilde{f}(x,y)\right) + H\left(\tilde{f}(x,y), f^\star(x,\cdot)\right)\right)^2$$

$$\leq 2\underbrace{\sum_{i=1}^{n} \mathop{\mathbb{E}}_{x\sim\mathcal{D}_i} H^2\left(f(x,\cdot),\tilde{f}(x,y)\right)}_{\text{(iii)}} + 2\underbrace{\sum_{i=1}^{n} \mathop{\mathbb{E}}_{x\sim\mathcal{D}_i} H^2\left(\tilde{f}(x,y), f^\star(x,\cdot)\right)}_{\text{(iv)}}.$$

For (iii), by the definition, we have $\tilde{f}(x,y) - f(x,y) \in [0,\epsilon]$ for all $x$, so

$$\text{(iii)} = \sum_{i=1}^{n} \mathop{\mathbb{E}}_{x\sim\mathcal{D}_i} H^2\left(f(x,\cdot),\tilde{f}(x,y)\right) \leq \sum_{i=1}^{n} \mathop{\mathbb{E}}_{x\sim\mathcal{D}_i} 2\int_y \left|f(x,y) - \tilde{f}(x,y)\right| \mathrm{d}y \leq 2n\epsilon|\mathcal{Y}|.$$

For (iv), we apply Lemma E.2 with $f_1 = f^\star$ and $f_2 = \tilde{f}$ (thus $s = 1 + \epsilon|\mathcal{Y}|$) and get

$$\text{(iv)} = n\epsilon|\mathcal{Y}| - 2\sum_{i=1}^{n} \log \mathop{\mathbb{E}}_{x,y\sim f^\star(x,\cdot)} \exp\left(-\frac{1}{2}\log\left(f^\star(x,y)/\tilde{f}(x,y)\right)\right)$$

$$= n\epsilon|\mathcal{Y}| - 2\sum_{i=1}^{n} \log \mathop{\mathbb{E}}_{x,y\sim\mathcal{D}_i} \exp\left(-\frac{1}{2}\log\left(f^\star(x,y)/\tilde{f}(x,y)\right)\right)$$

$$= n\epsilon|\mathcal{Y}| - 2\log \mathop{\mathbb{E}}_{x,y\sim\mathcal{D}'} \left[\exp\left(\sum_{i=1}^{n} -\frac{1}{2}\log\left(f^\star(x,y)/\tilde{f}(x,y)\right)\right)\Big|D\right]$$

$$= n\epsilon|\mathcal{Y}| + 2\cdot\text{(i)}.$$

By plugging (iii) and (iv) back we get

$$\sum_{i=1}^{n} \mathop{\mathbb{E}}_{x\sim\mathcal{D}_i} H^2\left(f(x,\cdot), f^\star(x,\cdot)\right) \leq 6n\epsilon|\mathcal{Y}| + 4\cdot\text{(i)}.$$

Notice that (i) $\leq$ (ii), so we complete the proof by plugging (ii) into the above. $\square$

We first state the MLE generalization result for finite $\mathcal{F}$.

**Theorem E.4.** *Suppose $\mathcal{F}$ is finite. Fix any $\delta \in (0,1)$, set $\beta = \log(|\mathcal{F}|/\delta)$ and define*

$$\widehat{\mathcal{F}} = \left\{f \in \mathcal{F} : \sum_{i=1}^{n} \log f(x_i, y_i) \geq \max_{\tilde{f}\in\mathcal{F}} \sum_{i=1}^{n} \tilde{f}(x_i, y_i) - 4\beta\right\}.$$

*Then w.p. at least $1 - \delta$, the following holds:*

*(1) The true distribution is in the version space, i.e., $f^\star \in \widehat{\mathcal{F}}$.*

*(2) Any function in the version space is close to the ground truth data-generating distribution, i.e., for all $f \in \widehat{\mathcal{F}}$*

$$\sum_{i=1}^{n} \mathbb{E}_{x\sim\mathcal{D}_i}\left[H^2(f(x,\cdot) \parallel f^\star(x,\cdot))\right] \leq 22\beta.$$

*Proof.* These two claims follow from Lemma E.3 with $\epsilon = 0$, and so $N_{[]}(\epsilon, \mathcal{F}, \|\cdot\|_\infty) = |\mathcal{F}|$. For (1), apply Eq. (5) to $f = \widehat{f}_{\text{MLE}}$ to see that $f^\star \in \widehat{\mathcal{F}}$. For (2), apply Eq. (4) and note that the sum term is at most $4\beta$. Thus, the right hand side of Eq. (4) is at most $(6 + 8 + 8)\beta = 22\beta$. $\square$

We now state the result for infinite $\mathcal{F}$ using bracketing entropy.

**Theorem E.5.** *Fix any $\delta \in (0,1)$, set $\beta = \log(N_{[]}((n|\mathcal{Y}|)^{-1}, \mathcal{F}, \|\cdot\|_\infty)/\delta)$ and define*

$$\widehat{\mathcal{F}} = \left\{ f \in \mathcal{F} : \sum_{i=1}^{n} \log f(x_i, y_i) \geq \max_{\widetilde{f} \in \mathcal{F}} \sum_{i=1}^{n} \widetilde{f}(x_i, y_i) - 7\beta \right\}.$$

*Then w.p. at least $1 - \delta$, the following holds:*

*(1) The true distribution is in the version space, i.e., $f^\star \in \widehat{\mathcal{F}}$.*

*(2) Any function in the version space is close to the ground truth data-generating distribution, i.e., for all $f \in \widehat{\mathcal{F}}$*

$$\sum_{i=1}^{n} \mathbb{E}_{x \sim \mathcal{D}_i} \left[ H^2(f(x, \cdot) \parallel f^\star(x, \cdot)) \right] \leq 28\beta.$$

*Proof.* These two claims follow from Lemma E.3 with $\epsilon = 1/n|\mathcal{Y}|$. For (1), apply Eq. (5) to $f = \widehat{f}_{\mathrm{MLE}}$ to see that $f^\star \in \widehat{\mathcal{F}}$. For (2), apply Eq. (4) and note that the sum term is at most $7\beta$. Thus, the right hand side of Eq. (5) is at most $(6 + 14 + 8)\beta = 28\beta$. $\qquad\square$

# F   Confidence set construction with general function class

In this section, we extend the confidence set construction of O-DISCO and P-DISCO to general $\mathcal{F}$, which can be infinite. Our procedure constructs the confidence set by performing the thresholding scheme on an $\varepsilon$-net of $\mathcal{F}$. While constructing an $\varepsilon$-net for $\mathcal{F}$ is admittedly a computationally hard procedure, this is still information theoretically possible and our focus in O-DISCO and P-DISCO is to show that distributional RL information-theoretically leads to small-loss bounds.

We first define some notations. Let $\mathcal{F}^\downarrow$ and $\mathcal{F}^\uparrow$ denote a lower and upper $\varepsilon$-bracketing of $\mathcal{F}$, i.e., for any $f \in \mathcal{F}$, there exists an $\varepsilon$-bracket $[f^\downarrow, f^\uparrow]$ such that for all $h$, $f_h^\downarrow \leq f_h \leq f_h^\uparrow$ with $f^\downarrow \in \mathcal{F}^\downarrow, f^\uparrow \in \mathcal{F}^\uparrow$. Recall that a lower bracket $g \in \mathcal{F}^\downarrow$ may not be a valid distribution, but since elements of $\mathcal{F}$ map to non-negative values, we can assume $g$ has non-negative entires as well. Also, we have $\alpha_h^g(x, a) := \int g_h(z \mid x, a) \geq 1 - \varepsilon$, so for $\varepsilon$ small enough, $g$ is normalizable. Hence, define $\widetilde{g}(z \mid x, a) = \alpha_h^g(x, a)^{-1} g(z \mid x, a)$ as the normalized version, which is a valid distribution that we can sample from.

Now, consider any martingale $\{x_{h,i}, a_{h,i}, c_{h,i}\}_{i \in [n], h \in [H]}$, which could be the online data up to episode $k$ or the offline data (consisting of $N$ i.i.d. samples). We define the MLE with respect to a lower bracket element as follows. For any $h \in [H], g \in \mathcal{F}^\downarrow, \pi \in \Pi$, sample $y_{h,i}^{g,\pi} \sim \widetilde{g}_{h+1}(x'_{h,i}, \pi(x'_{h,i}))$, and $z_{h,i}^{g,\pi} = c_{h,i} + y_{h,i}^{g,\pi}$, define the MLE solution for $(g, \pi)$ at time $h$ as,

$$\mathrm{MLE}_h^{g,\pi} = \arg\max_{f \in \mathcal{F}} \sum_{i=1}^{n} \log f_h(z_{h,i}^{g,\pi} \mid x_{h,i}, a_{h,i}).$$

Also, define the version space with respect to the above MLE as,

$$\mathcal{F}_{g,\pi,h} = \left\{ f \in \mathcal{F} : \sum_{i=1}^{n} \log f_h(z_{h,i}^{g,\pi} \mid x_{h,i}, a_{h,i}) \geq \sum_{i=1}^{n} \log \mathrm{MLE}_h^{g,\pi}(z_{h,i}^{g,\pi} \mid x_{h,i}, a_{h,i}) - \beta \right\}.$$

We now prove a key result that implies that $\mathcal{T}_h^\pi f_{h+1}^\downarrow$ falls into the confidence set $\mathcal{F}_{f^\downarrow,\pi,h}$.

**Theorem F.1.** *For any $\delta \in (0,1)$ and suppose $n \geq 2$. Then, w.p. at least $1 - \delta$, for any $h \in [H], g \in \mathcal{F}, f^\downarrow \in \mathcal{F}^\downarrow, \pi \in \Pi$, we have*

$$\sum_{i=1}^{n} \log g_h(z_{h,i}^{f^\downarrow,\pi} \mid x_{h,i}, a_{h,i}) - \log \mathcal{T}_h^\pi f_{h+1}^\downarrow(z_{h,i}^{f^\downarrow,\pi} \mid x_{h,i}, a_{h,i}) \leq \log(e^4 N_{[]}(n^{-1}, \mathcal{F}, \|\cdot\|_\infty)^2 |\Pi|/\delta).$$

*where $z_{h,i}^{f^\downarrow,\pi} = c_{h,i} + y_{h,i}^{f^\downarrow,\pi}$ and $y_{h,i}^{f^\downarrow,\pi} \sim \widetilde{f}_{h+1}^\downarrow(\cdot \mid x'_{h,i}, \pi_{h+1}(x'_{h,i}))$.*

*Proof of Theorem F.1.* Consider a $\varepsilon$-bracketing of $\mathcal{F}$ where $\varepsilon \leq 1/n \leq 1/2$; we will study each element and conclude with a union bound. For any lower bracket $l$ and upper bracket $u$ in the bracketing (note $l, u$ need not correspond to the same bracket). Recall that $\alpha_{h+1}^l(x, a) := \int l_{h+1}(z \mid x, a)$, so we have $1 - \varepsilon \leq \alpha_{h+1}^l \leq 1$ since $l$ is a lower $\varepsilon$-bracket of distributions. Therefore, we have

$$\mathbb{E}\left[\exp \sum_{i=1}^n \log\left(\frac{u_h(z_{h,i}^{l,\pi} \mid x_{h,i}, a_{h,i})}{\mathcal{T}_h^\pi l_{h+1}(z_{h,i}^{l,\pi} \mid x_{h,i}, a_{h,i})}\right)\right] = \prod_{i=1}^n \mathbb{E}_{\nu_{h,i}}\left[\frac{u_h(z_{h,i}^{l,\pi} \mid x_{h,i}, a_{h,i})}{\mathcal{T}_h^\pi l_{h+1}(z_{h,i}^{l,\pi} \mid x_{h,i}, a_{h,i})}\right],$$

where $\nu_{h,i}$ is the distribution of data from $i$-th round and time $h$. Note that $\nu_{h,i}(x, a, c, x') = d_{h,i}(x, a)C_h(c \mid x, a)P_h(x' \mid x, a)$ for some distribution $d_{h,i}(x, a)$. Now focus on each $i$, so for all $i$, we have

$$\mathbb{E}_{\nu_{h,i}}\left[\frac{u_h(z_{h,i}^{l,\pi} \mid x_{h,i}, a_{h,i})}{\mathcal{T}_h^\pi l_{h+1}(z_{h,i}^{l,\pi} \mid x_{h,i}, a_{h,i})}\right]$$

$$= \int_{x,a,c,x',y} \nu_{h,i}(x, a, c, x')\widetilde{l}_{h+1}(y \mid x', \pi(x')) \frac{u_h(c + y \mid x, a)}{\int_{c,x'} \nu_{h,i}(c, x' \mid x, a)l_{h+1}(y \mid x', \pi(x'))}$$

$$= \int_{x,a,z} d_{h,i}(x, a) \int_z u_h(z \mid x, a)$$

$$\times \int_{c,x'} \nu_{h,i}(c, x' \mid x, a)\widetilde{l}_{h+1}(z - c \mid x', \pi(x')) \frac{1}{\int_{c,x'} \nu_{h,i}(c, x' \mid x, a)l_{h+1}(z - c \mid x', \pi(x'))}$$

$$= \int_{x,a,z} d_{h,i}(x, a) \int_z u_h(z \mid x, a)\alpha_{h+1}^l(x, a)^{-1}$$

$$\leq \frac{1 + \varepsilon}{1 - \varepsilon} = 1 + \frac{2\varepsilon}{1 - \varepsilon} \leq 1 + \frac{4}{n}.$$

Therefore,

$$\mathbb{E}\left[\exp \sum_{i=1}^n \log\left(\frac{u_h(z_{h,i}^{l,\pi} \mid x_{h,i}, a_{h,i})}{\mathcal{T}_h^\pi l_{h+1}(z_{h,i}^{l,\pi} \mid x_{h,i}, a_{h,i})}\right)\right] \leq (1 + 4/n)^n \leq e^4.$$

Thus, by Markov's inequality, w.p. at least $1 - \delta$, we have

$$\sum_{i=1}^n \log\left(\frac{u_h(z_{h,i}^{l,\pi} \mid x_{h,i}, a_{h,i})}{\mathcal{T}_h^\pi l_{h+1}(z_{h,i}^{l,\pi} \mid x_{h,i}, a_{h,i})}\right) \leq \ln(e^4/\delta).$$

To conclude, apply union bound to get this result for all brackets. $\qquad\square$

For the remainder of this section, we assume the policy class $\Pi$ is finite. However, it is possible to extend our results using policy covers in the Hamming distance; in that case, $\log|\Pi|$ would be replaced by the log covering number or entropy integral of $\Pi$ [as in Zhou et al., 2023, Kallus et al., 2022]. We note that for the *online* case, we rely on the assumption that for any $f \in \mathcal{F}$ we have $\pi^f \in \Pi$, where recall that $\pi_h^f(x) = \arg\min_a \bar{f}_h(x, a)$. This is because $\mathcal{T}^{\star,D}$ is not a contraction so we cannot operate with $\mathcal{T}^{\star,D}$ directly and instead operate with $\mathcal{T}^{\pi^f,D}$. We highlight that this assumption is automatically satisfied in tabular MDPs, since the whole policy space is finite, and $\log|\Pi| = \mathcal{O}(X \log(A))$ is lower order compared to log of the bracketing entropy of $\mathcal{F}_{tab}$, which is $\mathcal{O}(X^2 A^2)$. In contrast, in non-distributional methods such as GOLF, the regular Bellman optimality operator is a contraction so standard Lipschitz arguments for covering go through. We note that it is also possible to construct covers of $\mathcal{F}$ in the Hellinger distance, but the metric entropy of $\mathcal{F}_{tab}$ seems to be on the same order as its bracketing entropy.

We now describe the version space construction for general $\mathcal{F}$, first for the online setting. Fix any $k$, and define the set

$$\mathcal{F}_{f^\downarrow,\pi,h} = \left\{f \in \mathcal{F} : \sum_{i=1}^k \log f_h(z_{h,i}^{f^\downarrow,\pi} \mid x_{h,i}, a_{h,i}) \geq \sum_{i=1}^k \log \mathrm{MLE}_h^{f^\downarrow,\pi}(z_{h,i}^{f^\downarrow,\pi} \mid x_{h,i}, a_{h,i}) - \beta\right\}$$

Then, construct the version space as

$$\mathcal{F}_k = \left\{f \in \mathcal{F} : f_h \in \mathcal{F}_{f^\downarrow,\pi^f,h}, \forall h \in [H]\right\}.$$

**Theorem F.2.** *Fix any $\delta \in (0, 1)$ and suppose [Assumption 5.1]. Set $\beta = \log(KH \cdot N_{[]}(K^{-1}, \mathcal{F}, \| \cdot \|_\infty) |\Pi|/\delta)$. Then, w.p. at least $1 - \delta$, the following holds:*

*(1) The optimal cost distribution is in the version space, i.e., $Z^\star \in \mathcal{F}_k$.*

*(2) For all $f \in \mathcal{F}_k$ and $h \in [H]$,*

$$\sum_{i=1}^{k} \mathbb{E}_{\pi^i} \left[ H^2(f_h(x_h, a_h) \parallel \mathcal{T}_h^{\star, D} f_{h+1}(x_h, a_h)) \right] \leq 60\beta.$$

*Proof.* First, we want to verify that $Z^\star \in \mathcal{F}_k$. Let $f^\downarrow$ be the lower bracket of $Z^\star$ and set $g = \text{MLE}_h^{f^\downarrow, \pi^\star} \in \mathcal{F}$; note $\pi^\star = \pi^{Z^\star}$. By [Theorem F.1], we have $\sum_{i=1}^{k} \log \text{MLE}_h^{f^\downarrow, \pi^\star}(z_{h,i}^{f^\downarrow, \pi^\star} \mid x_{h,i}, a_{h,i}) - \log \mathcal{T}_h^{\pi^\star, D} f_{h+1}^\downarrow(z_{h,i}^{f^\downarrow, \pi^\star} \mid x_{h,i}, a_{h,i}) \leq \mathcal{O}(\beta)$. Therefore, noting that $Z_h^\star = \mathcal{T}_h^{\pi^\star, D} Z_{h+1}^\star \geq \mathcal{T}_h^{\pi^\star, D} f_{h+1}^\downarrow$ shows that $Z_h^\star \in \mathcal{F}_{f^\downarrow, \pi^\star, h}$ for every $h$, implying that $Z^\star \in \mathcal{F}_k$.

For the second claim, fix any $f \in \mathcal{F}_k$ and $h \in [H]$. Then,

$$\sum_{i=1}^{k} \mathbb{E}_{\pi^i} \left[ H^2(f_h(x_h, a_h) \parallel \mathcal{T}_h^{\star, D} f_{h+1}(x_h, a_h)) \right]$$

$$= \sum_{i=1}^{k} \mathbb{E}_{\pi^i} \left[ H^2(f_h(x_h, a_h) \parallel \mathcal{T}_h^{\pi^f, D} f_{h+1}(x_h, a_h)) \right]$$

$$\leq 2 \sum_{i=1}^{k} \mathbb{E}_{\pi^i} \left[ H^2(f_h(x_h, a_h) \parallel \mathcal{T}_h^{\pi^f, D} \widetilde{f}_{h+1}^\downarrow(x_h, a_h)) + H^2(\mathcal{T}_h^{\pi^f, D} \widetilde{f}_{h+1}^\downarrow(x_h, a_h) \parallel \mathcal{T}_h^{\pi^f, D} f_{h+1}(x_h, a_h)) \right]$$

$$\leq 2(28\beta + 3k\varepsilon).$$

The $\beta$ comes from [Theorem E.5], and for $\varepsilon$, we used the fact that $H^2 \leq H \leq TV$, and

$$\sum_{i=1}^{k} \mathbb{E}_{\pi^i} \left[ TV(\mathcal{T}_h^{\pi^f, D} \widetilde{f}_{h+1}^\downarrow(x_h, a_h) \parallel \mathcal{T}_h^{\pi^f, D} f_{h+1}(x_h, a_h)) \right]$$

$$= \sum_{i=1}^{k} \mathbb{E}_{\pi^i} \int_z \left| \mathcal{T}_h^{\pi^f, D} \widetilde{f}_{h+1}^\downarrow(z \mid x_h, a_h) - \mathcal{T}_h^{\pi^f, D} f_{h+1}(z \mid x_h, a_h) \right|$$

$$= \sum_{i=1}^{k} \mathbb{E}_{\pi^i} \int_z \sum_{c, x'} \nu(c, x' \mid x_h, a_h) \left| \widetilde{f}_{h+1}^\downarrow(z - c \mid x', \pi^f(x')) - f_{h+1}(z - c \mid x', \pi^f(x')) \right|$$

$$\leq \sum_{i=1}^{k} 3\varepsilon = 3k\varepsilon,$$

since for any $x, a$, we have $\int_z \left| \widetilde{f}_{h+1}^\downarrow(z \mid x, a) - f_{h+1}(z \mid x, a) \right| \leq 3\varepsilon$. There are two cases. If $\widetilde{f}_{h+1}^\downarrow(z \mid x, a) \geq f_{h+1}(z \mid x, a)$, then $\widetilde{f}_{h+1}^\downarrow(z \mid x, a) - f_{h+1}(z \mid x, a) \leq (1 - \varepsilon)^{-1} f_{h+1}^\downarrow(z \mid x, a) - f_{h+1}(z \mid x, a) \leq 2\varepsilon f_{h+1}(z \mid x, a)$ since $(1 - \varepsilon)^{-1} \leq 1 + 2\varepsilon$. If $\widetilde{f}_{h+1}^\downarrow(z \mid x, a) < f_{h+1}(z \mid x, a)$, then $f_{h+1}(z \mid x, a) - \widetilde{f}_{h+1}^\downarrow(z \mid x, a) \leq f_{h+1}(z \mid x, a) - f_{h+1}^\downarrow(z \mid x, a) \leq \varepsilon$. Thus, $\int_z \max(2\varepsilon f_{h+1}(z \mid x, a), \varepsilon) \leq \int_z 2\varepsilon f_{h+1}(z \mid x, a) + \varepsilon = 3\varepsilon$. Thus, setting $\varepsilon = 1/K$ gives

$$\sum_{i=1}^{k} \mathbb{E}_{\pi^i} \left[ H^2(f_h(x_h, a_h) \parallel \mathcal{T}_h^{\star, D} f_{h+1}(x_h, a_h)) \right] \leq 59\beta.$$

$\square$

For the offline setting, fix any $\pi$ and define its general version space as,

$$\mathcal{F}_\pi = \left\{ f \in \mathcal{F} : f_h \in \mathcal{F}_{f^\downarrow, \pi, h}, \forall h \in [H] \right\}.$$

**Theorem F.3.** *Fix any* $\delta \in (0,1)$ *and suppose* Assumption 5.1*. Set* $\beta = \log(H|\Pi| \cdot N_{[]}((n|\mathcal{Y}|)^{-1}, \mathcal{F}, \|\cdot\|_\infty)/\delta)$*. Then, w.p. at least* $1 - \delta$*, the following holds for all policies* $\pi \in \Pi$*:*

*(1) The policy cost distribution is in the version space, i.e.,* $Z^\pi \in \mathcal{F}_\pi$*.*

*(2) Any function in the version space has bounded triangular discrimination with the ground truth data-generating distribution, i.e., for all* $f \in \mathcal{F}_\pi$ *and* $h \in [H]$*,*

$$\mathbb{E}_{\nu_h}\left[H^2(f_h(x_h, a_h) \| \mathcal{T}_h^{\pi, D} f_{h+1}(x_h, a_h))\right] \leq 60\beta N^{-1}.$$

*Proof.* The proof is the same as in Theorem F.2, but instead of $\pi^f$, we fix any $\pi$. $\qquad\square$

## G    The $\ell_p$ distributional eluder dimension

Let $\mathcal{S}$ denote any input space (for example, we will later instantiate $\mathcal{S} = \mathcal{X}$ or $\mathcal{S} = \mathcal{X} \times \mathcal{A}$). Let $\Psi$ denote a set of functions mapping from $\mathcal{S} \to \mathbb{R}$. Let $\mathcal{D}$ be a set of distributions on $\mathcal{S}$.

Recall the definition of $\varepsilon$-independent sequence (of distributions) from Jin et al. [2021a].

**Definition G.1** ($\ell_2$-independent sequence)**.** A distribution $\nu \in \mathcal{D}$ is $(\varepsilon, \ell_2)$-independent of a sequence $\{d^{(1)}, \ldots, d^{(n)}\} \subset \mathcal{D}$ if there exists $\psi \in \Psi$ such that $|\mathbb{E}_\nu \psi| > \varepsilon$ and also $\sqrt{\sum_{i=1}^n (\mathbb{E}_{d^{(i)}} \psi)^2} \leq \varepsilon$.

Note that the definition is on sequences of distributions, which generalizes the original definition on sequences of points from Russo and Van Roy [2013].

We now generalize the above definition for the general $\ell_p$ norm.

**Definition G.2** ($\ell_p$-independent sequence)**.** A distribution $\nu \in \mathcal{D}$ is $(\varepsilon, \ell_p)$-independent of a sequence $\{d^{(1)}, \ldots, d^{(n)}\} \subset \mathcal{D}$ if there exists $\psi \in \Psi$ such that $|\mathbb{E}_\nu \psi| > \varepsilon$ and also $\sum_{i=1}^n |\mathbb{E}_{d^{(i)}} \psi|^p \leq \varepsilon^p$.

Using the definition of independent sequences established so far, we define the $\ell_p$ distributional eluder dimension.

**Definition G.3** ($\ell_p$-distributional eluder dimension)**.** For any $p$, define the $\ell_p$-distributional eluder dimension (denoted by $\mathrm{DE}_p(\Psi, \mathcal{D}, \varepsilon)$) as the length of the longest sequence $\{d^{(1)}, \ldots, d^{(d)}\} \subset \mathcal{D}$ such that there exists $\varepsilon' \geq \varepsilon$, such that for all $t \in [d]$, $d^{(t)}$ is $(\varepsilon', \ell_p)$-independent of $d^{(1)}, \ldots, d^{(t-1)}$.

Of particular interest to us is the $\ell_1$ case. We show that the $\ell_1$ eluder dimension is dominated by the $\ell_2$ eluder dimension of Jin et al. [2021a].

**Lemma 5.4.** *For any* $\Psi, \mathcal{D}$ *and* $\varepsilon > 0$*, we have* $\mathrm{DE}_1(\Psi, \mathcal{D}, \varepsilon) \leq \mathrm{DE}_2(\Psi, \mathcal{D}, \varepsilon)$*.*

*Proof.* Since $\sqrt{\sum_i x_i^2} \leq \sum_i |x_i|$, we have that any witness (long independent sequence) for $\ell_1$ is also a witness for $\ell_2$. So, the maximum length of the $\ell_2$ witnesses is longer than the $\ell_1$ witnesses. Liu et al. [2022, Proposition 19] obtains an analogous result for the non-distributional eluder dimension of Russo and Van Roy [2013]. $\qquad\square$

We now prove the key pigeonhole result for the $\ell_1$ distributional eluder dimension.

**Theorem 5.3.** *Let* $C := \sup_{d \in \mathcal{D}, f \in \Psi} |\mathbb{E}_d f|$ *be the envelope. Fix any* $K \in \mathbb{N}$ *and sequences* $f^{(1)}, \ldots, f^{(K)} \subseteq \Psi$*,* $d^{(1)}, \ldots, d^{(K)} \subseteq \mathcal{D}$*. Let* $\beta$ *be a constant such that for all* $k \in [K]$*, we have,* $\sum_{i=1}^{k-1} |\mathbb{E}_{d^{(i)}} f^{(k)}| \leq \beta$*. Then, for all* $k \in [K]$*, we have*

$$\sum_{t=1}^k \left|\mathbb{E}_{d^{(t)}} f^{(t)}\right| \leq \inf_{0 < \varepsilon \leq 1} \{\mathrm{DE}_1(\Psi, \mathcal{D}, \varepsilon)(2C + \beta \log(C/\varepsilon)) + k\varepsilon\}.$$

*Proof.* For any $\Gamma \subset \mathcal{D}$, $\nu \in \mathcal{D}$, and $0 < \varepsilon \leq 1$, let $L(\nu, \Gamma, \varepsilon)$ denote the number of disjoint subsets of $\Gamma$ such that each subset is $\varepsilon$-dependent of $\nu$, *i.e.*, for all such disjoint subsets of $\Gamma$, it is not the case that $\nu$ is $(\varepsilon, \ell_1)$-independent of each subset.

**Fact 1: For any $\varepsilon$, if $\left|\mathbb{E}_{d^{(k)}} f^{(k)}\right| > \varepsilon$ for some $k \in [K]$, then $L(d^{(k)}, d^{(1:k-1)}, \varepsilon) < \beta/\varepsilon$.**
By definition of $L := L(d^{(k)}, d^{(1:k-1)}, \varepsilon)$, there exist disjoint subsequences $\mathfrak{G}^{(1)}, \ldots, \mathfrak{G}^{(L)}$ of $d^{(1:k-1)}$ such that each subsequence $\mathfrak{G}^{(i)}$ satisfies $\sum_{d \in \mathfrak{G}^{(i)}} \left|\mathbb{E}_d f^{(k)}\right| > \varepsilon$. Therefore, summing over all subsequences, we have $L\varepsilon < \sum_{i=1}^{k-1} \left|\mathbb{E}_{d^{(i)}} f^{(k)}\right| \leq \beta$, where the $\beta$ inequality comes from the premise. This proves Fact 1.

**Fact 2: For any $\varepsilon$ and any sequence $\left\{\nu^{(1)}, \ldots, \nu^{(\kappa)}\right\} \subset \mathcal{D}$, there exists $j \in [\kappa]$ such that $L(\nu^{(j)}, \nu^{(1:j-1)}, \varepsilon) \geq J := \lfloor (\kappa - 1) / \mathrm{DE}_1(\Psi, \mathcal{D}, \varepsilon) \rfloor$.**
If $J = 0$, the claim is vacuously true. Otherwise, consider the following algorithm for finding the $j$:

Step 1) Initialize $\mathfrak{G}^{(1)} = [\nu^{(1)}], \ldots, \mathfrak{G}^{(J)} = [\nu^{(J)}]$ and let $j = J + 1$.

Step 2) If $\nu^{(j)}$ is $\varepsilon$-dependent on all of $\mathfrak{G}^{(i)}, i \in [J]$, then the claim is proven and terminate.

Step 3) Otherwise, there exists some $\mathfrak{G}^{(i)}, i \in [J]$ such that $\nu^{(j)}$ is $\varepsilon$-independent of it. Append $\nu^{(j)}$ to $\mathfrak{G}^{(i)}$, *i.e.*, $\mathfrak{G}^{(i)} = \mathfrak{G}^{(i)} + [\nu^{(j)}]$. Increment $j = j + 1$ and go back to Step 2.

Hence, we need to argue this process terminates at Step 2 before $j$ gets to $\kappa + 1$. We prove this by contradiction: assume $j$ gets to $\kappa + 1$. Let $i \in [J]$ be such that $\mathfrak{G}^{(i)}$ has the most elements (break ties arbitrarily). Since $\kappa = \sum_{i=1}^{J} \left|\mathfrak{G}^{(i)}\right| \leq J\left|\mathfrak{G}^{(i)}\right|$, we have that $\left|\mathfrak{G}^{(i)}\right| \geq \kappa/J \geq \frac{\kappa}{\kappa-1} \mathrm{DE}_1(\Psi, \mathcal{D}, \varepsilon) > \mathrm{DE}_1(\Psi, \mathcal{D}, \varepsilon)$, where we've also used the definition of $J$. By construction, $\mathfrak{G}^{(i)}$ is an $\varepsilon$-eluder sequence, *i.e.*, it is a sequence such that each element is $\varepsilon$-independent of its predecessors. However, this is a contradiction because its size is greater than $\mathrm{DE}_1(\Psi, \mathcal{D}, \varepsilon)$. Therefore, this process terminates at Step 2 for some $j$, which is the witness for proving Fact 2.

**Fact 3: For any $\varepsilon$ and $k \in [K]$, we have $\sum_{t=1}^{k} \mathbb{I}\left[\left|\mathbb{E}_{d^{(t)}} f^{(t)}\right| > \varepsilon\right] \leq \left(\beta\varepsilon^{-1} + 1\right) \mathrm{DE}_1(\Psi, \mathcal{D}, \varepsilon) + 1$.**
Fix any $\varepsilon$ and $k \in [K]$. Let $\left\{d^{(i_1)}, \ldots, d^{(i_\kappa)}\right\}$ be all the elements of $d^{(1:k)}$ such that $\mathbb{E}_{d^{(t)}} f^{(t)} > \varepsilon$ for $t = i_1, \ldots, i_\kappa$. By Fact 2, there exists $j \in [\kappa]$ such that $L(d^{(i_j)}, d^{(i_{1:j-1})}, \varepsilon) \geq \lfloor (\kappa - 1) / \mathrm{DE}_1(\Psi, \mathcal{D}, \varepsilon) \rfloor$. By Fact 1, we have $L(d^{(i_j)}, d^{(1:i_j)}, \varepsilon) \leq \beta/\varepsilon$. Finally notice that $L(d^{(i_j)}, d^{(i_{1:j-1})}, \varepsilon) \leq L(d^{(i_j)}, d^{(1:i_j)}, \varepsilon)$ since adding more elements can only create more $\varepsilon$-dependent-of-$\nu$ disjoint subsets. Thus, combining these inequalities, we have $\lfloor (\kappa - 1) / \mathrm{DE}_1(\Psi, \mathcal{D}, \varepsilon) \rfloor < \beta/\varepsilon$. This implies $\kappa \leq \left(\beta\varepsilon^{-1} + 1\right) \mathrm{DE}_1(\Psi, \mathcal{D}, \varepsilon) + 1$, which proves Fact 3.

**Finishing the proof**
Fix any $k \in [K]$ and $\omega > 0$. We have

$$\sum_{t=1}^{k} \left|\mathbb{E}_{d^{(t)}} f^{(t)}\right| = \sum_{t=1}^{k} \int_0^C \mathbb{I}\left[\left|\mathbb{E}_{d^{(t)}} f^{(t)}\right| > y\right] \mathrm{d}y$$

$$\leq k\omega + \sum_{t=1}^{k} \int_\omega^C \mathbb{I}\left[\left|\mathbb{E}_{d^{(t)}} f^{(t)}\right| > y\right] \mathrm{d}y$$

$$= k\omega + \int_\omega^C \sum_{t=1}^{k} \mathbb{I}\left[\left|\mathbb{E}_{d^{(t)}} f^{(t)}\right| > y\right] \mathrm{d}y$$

$$\leq k\omega + \int_\omega^C \left\{(\beta/y + 1) \mathrm{DE}_1(\Psi, \mathcal{D}, y) + 1\right\} \mathrm{d}y \qquad \text{(Fact 3)}$$

$$\leq k\omega + \int_\omega^C \left\{(\beta/y + 1) \mathrm{DE}_1(\Psi, \mathcal{D}, \omega) + 1\right\} \mathrm{d}y \qquad \text{(Monotonicity of } \mathrm{DE}_1)$$

$$\leq k\omega + (d+1)C + d\beta \log(C/\omega). \qquad (d := \mathrm{DE}_1(\Psi, \mathcal{D}, \omega))$$

This completes the proof. $\qquad \square$

## G.1 Bounding V-type $\ell_2$ eluder dimension in low-rank MDPs

**Theorem G.4** (Bound of $\ell_2$ distributional eluder for low-rank MDPs). *Suppose the MDP is a low-rank MDP. Let $\Psi \subset \mathcal{X} \to [0,1]$ be any class of functions mapping $\mathcal{X}$ to $[0,1]$. Suppose*

$\mathcal{D} = \{x \mapsto d_h^\pi(x) : \pi \in \Pi\}$ *for some* $h \in [H]$. *Then, we have*

$$\mathrm{DE}_2(\Psi, \mathcal{D}, \varepsilon) \leq \mathcal{O}(d \log(d/\varepsilon)). \tag{7}$$

*Proof.* If $h = 1$, then $\mathcal{D}$ is a singleton. Hence, $\mathrm{DE}_2(\Psi, \mathcal{D}, \varepsilon) \leq 1$. Hence, suppose $h \geq 2$; set $h := h - 1$ and we will focus on $d_{h+1}^\pi$ in the remainder. Suppose $\left\{d^{(k)}, f^{(k)}\right\}_{k \in [T]}$ is any sequence such for all $k \in [T]$, we have that $(d^{(k)}, f^{(k)})$ is $(\varepsilon, \ell_2)$-independent of its predecessors. For any $k$, set $\Sigma_k = \sum_{i=1}^{k-1} \mathbb{E}_{d^{(i)}}[\phi_h^\star(x_h, a_h)] \mathbb{E}_{d^{(i)}}[\phi_h^\star(x_h, a_h)]^\top + \lambda I$. Then, we have

$$
\begin{aligned}
\mathbb{E}_{d^{(k)}} f^{(k)}(x_{h+1}) &= \mathbb{E}_{d^{(k)}} \int_{x_h} \phi_h^\star(x_h, a_h)^\top \mathrm{d}\mu_h^\star(x_{h+1}) f^{(k)}(x_{h+1}) \\
&= \mathbb{E}_{d^{(k)}} \phi_h^\star(x_h, a_h)^\top \int_{x_{h+1}} f^{(k)}(x_{h+1}) \mathrm{d}\mu_h^\star(x_{h+1}). \\
&\leq \|\mathbb{E}_{d^{(k)}} \phi_h^\star(x_h, a_h)\|_{\Sigma_k^{-1}} \|\int_{x_{h+1}} f^{(k)}(x_{h+1}) \mathrm{d}\mu_h^\star(x_{h+1})\|_{\Sigma_k}.
\end{aligned}
$$

Focusing on the second term,

$$\|\int_{x_{h+1}} f^{(k)}(x_{h+1}) \mathrm{d}\mu_h^\star(x_{h+1})\|_{\Sigma_k}^2 = \sum_{i=1}^{k-1} \Big( \mathbb{E}_{d^{(i)}} \Big[ f^{(k)}(x_{h+1}) \Big] \Big)^2 + \lambda d$$

Thus, we have shown that

$$\mathbb{E}_{d^{(k)}} f^{(k)}(x_{h+1}) \leq \|\mathbb{E}_{d^{(k)}} \phi_h^\star(x_h, a_h)\|_{\Sigma_k^{-1}} \sqrt{\sum_{i=1}^{k-1} \big( \mathbb{E}_{d^{(i)}} \big[ f^{(k)}(x_{h+1}) \big] \big)^2 + \lambda d}.$$

Then, by the independent sequence assumption, we have

$$
\begin{aligned}
T\varepsilon &< \sum_{k=1}^T \mathbb{E}_{d^{(k)}} f^{(k)}(x_{h+1}) \leq \sum_{k=1}^T \|\mathbb{E}_{d^{(k)}} \phi_h^\star(x_h, a_h)\|_{\Sigma_k^{-1}} \sqrt{\sum_{i=1}^{k-1} \big( \mathbb{E}_{d^{(i)}} \big[ f^{(k)}(x_{h+1}) \big] \big)^2 + \lambda d} \\
&\leq \sum_{k=1}^T \|\mathbb{E}_{d^{(k)}} \phi_h^\star(x_h, a_h)\|_{\Sigma_k^{-1}} \Big( \varepsilon + \sqrt{\lambda d} \Big) && \big( \sqrt{\sum_{i=1}^{k-1} \big( \mathbb{E}_{d^{(i)}} \big[ f^{(k)}(x_{h+1}) \big] \big)^2} \leq \varepsilon \big) \\
&\leq 2\varepsilon \sum_{k=1}^T \|\mathbb{E}_{d^{(k)}} \phi_h^\star(x_h, a_h)\|_{\Sigma_k^{-1}} && (\lambda = \varepsilon^2/d) \\
&\leq 2\varepsilon \sqrt{T} \sqrt{\sum_{k=1}^T \|\mathbb{E}_{d^{(k)}} \phi_h^\star(x_h, a_h)\|_{\Sigma_k^{-1}}^2} \\
&\leq 2\varepsilon \sqrt{T} \sqrt{d \log(1 + T/d\lambda)} && \text{(elliptical potential)} \\
&\leq 2\varepsilon \sqrt{T} \sqrt{d \log(1 + T/\varepsilon^2)}. && (\lambda = \varepsilon^2/d)
\end{aligned}
$$

For a reference of the elliptical potential, see Uehara et al. [2021, Lemmas 19&20]. Rearranging, we have $\sqrt{T} < 2\sqrt{d \log(1 + T/\varepsilon^2)}$, which implies

$$T \leq 4d \log(1 + T/\varepsilon^2).$$

By applying Lemma G.5, we have $T \leq 24d \log(1 + 4d/\varepsilon^2)$. This concludes the proof. $\qquad \square$

**Lemma G.5.** *Let* $c_1, c_2 \geq 1$ *be constants. Let* $x \geq 0$ *be a solution to* $x \leq c_1 \log(1 + c_2 x)$. *Then, we necessarily have* $x \leq 6c_1 \log(1 + c_1 c_2)$.

*Proof.* Using change of variables $B = \frac{x}{c_1}$, we have the inequality is equivalent to $B \leq \log(1 + B \cdot c_1 c_2)$. Take exp of both sides to get $\exp(B) \leq \alpha B + 1$ where $\alpha = c_1 c_2$. From Step 3 of the proof of Russo and Van Roy [2013, Proposition 6], we have $B \leq \frac{e}{e-1} \frac{e}{e-1} (\log(1 + \alpha) + \log(e/(e-1))) \leq 3(\log(1 + c_1 c_2) + 1)$. Hence, $x \leq c_1 \cdot 3(\log(1 + c_1 c_2) + 1)$. $\qquad \square$

## G.2 Bounding Q-type $\ell_2$ eluder dimension in tabular MDPs

**Theorem G.6** (Bound of $\ell_2$ distributional eluder for tabular MDPs). *Suppose the MDP is a tabular MDP. Let $\Psi \subset \mathcal{X} \times \mathcal{A} \to [0,1]$ be any class of functions mapping $\mathcal{X} \times \mathcal{A}$ to $[0,1]$. Suppose $\mathcal{D}$ be any set of distributions. Then, we have*

$$\mathrm{DE}_2(\Psi, \mathcal{D}, \varepsilon) \leq \mathcal{O}(SA \log(SA/\varepsilon)). \tag{8}$$

*Proof.* Suppose $\left\{d^{(k)}, f^{(k)}\right\}_{k \in [T]}$ is any sequence such for all $k \in [T]$, we have that $(d^{(k)}, f^{(k)})$ is $(\varepsilon, \ell_2)$-independent of its predecessors. Since the MDP is tabular, we can interpret $d^{(k)}, f^{(k)}$ as $SA$-dimensional vectors. For any $k$, set $\Sigma_k = \sum_{i=1}^{k-1} d^{(i)} (d^{(i)})^\top + \lambda I$. Then, we have

$$\mathbb{E}_{d^{(k)}} f^{(k)}(x, a) = (d^{(k)})^\top f^{(k)} \leq \|d^{(k)}\|_{\Sigma_k^{-1}} \|f^{(k)}\|_{\Sigma_k}.$$

Focusing on the second term, we have

$$\|f^{(k)}\|_{\Sigma_k}^2 = \sum_{i=1}^{k-1} \left((d^{(i)})^\top f^{(k)}\right)^2 + \lambda SA.$$

Thus, we have

$$T\varepsilon < \sum_{k=1}^{T} \mathbb{E}_{d^{(k)}} f^{(k)}(x, a) \leq \sum_{k=1}^{T} \|d^{(k)}\|_{\Sigma_k^{-1}} \sqrt{\sum_{i=1}^{k-1} \left(\mathbb{E}_{d^{(i)}}[f^{(k)}(x, a)]\right)^2 + \lambda SA}$$

$$\leq \sum_{k=1}^{T} \|d^{(k)}\|_{\Sigma_k^{-1}} \left(\varepsilon + \sqrt{\lambda SA}\right)$$

$$\leq 2\varepsilon \sum_{k=1}^{T} \|d^{(k)}\|_{\Sigma_k^{-1}} \qquad\qquad (\lambda = \varepsilon^2/SA)$$

$$\leq 2\varepsilon\sqrt{T} \sqrt{\sum_{k=1}^{T} \|d^{(k)}\|_{\Sigma_k^{-1}}^2}$$

$$\leq 2\varepsilon\sqrt{T}\sqrt{SA \log(1 + T/\varepsilon^2)}. \qquad\qquad \text{(elliptical potential)}$$

Rearranging, we have $\sqrt{T} < 2\sqrt{SA \log(1 + T/\varepsilon^2)}$, which implies $T \leq 4SA \log(1 + T/\varepsilon^2)$. Then by applying Lemma G.5, we have $T \leq 24SA \log(1 + {}^{4SA}/_{\varepsilon^2})$. This concludes the proof. $\qquad\square$

# H Proofs for Online RL

## H.1 Preliminary Lemmas

**Lemma H.1.** *For any policy $\pi$, conditional distribution $d$ and $h \in [H]$, we have*

$$\overline{\mathcal{T}_h^{\pi,D} d}(x,a) = \mathcal{T}_h^\pi \bar{d}(x,a),$$
$$\overline{\mathcal{T}_h^{\star,D} d}(x,a) = \mathcal{T}_h^\star \bar{d}(x,a).$$

*Proof.*

$$
\begin{aligned}
\overline{\mathcal{T}_h^{\pi,D} d}(x,a) &= \mathbb{E}_{y \sim \mathcal{T}_h^{\pi,D} d(x,a)}[y] \\
&= \mathbb{E}_{c \sim C_h(x,a), x' \sim P_h(x,a), a' \sim \pi_{h+1}(x'), y' \sim d(x',a')}[c + y'] \\
&= \bar{C}_h(x,a) + \mathbb{E}_{x' \sim P_h(x,a), a' \sim \pi_{h+1}(x'), y' \sim d(x',a')}[y'] \\
&= \bar{C}_h(x,a) + \mathbb{E}_{x' \sim P_h(x,a), a' \sim \pi_{h+1}(x')}[\bar{d}(x',a')] \\
&= \mathcal{T}_h^\pi \bar{d}(x,a).
\end{aligned}
$$

$$
\begin{aligned}
\overline{\mathcal{T}_h^{\star,D} d}(x,a) &= \mathbb{E}_{y \sim \mathcal{T}_h^{\star,D} d(x,a)}[y] \\
&= \mathbb{E}_{c \sim C_h(x,a), x' \sim P_h(x,a), a' = \arg\min_{\tilde{a}} \bar{d}(x',\tilde{a}), y' \sim d(x',a')}[c + y'] \\
&= \bar{C}_h(x,a) + \mathbb{E}_{x' \sim P_h(x,a), a' = \arg\min_{\tilde{a}} \bar{d}(x',\tilde{a}), y' \sim d(x',a')}[y'] \\
&= \bar{C}_h(x,a) + \mathbb{E}_{x' \sim P_h(x,a), a' = \arg\min_{\tilde{a}} \bar{d}(x',\tilde{a})}[\bar{d}(x',a')] \\
&= \bar{C}_h(x,a) + \mathbb{E}_{x' \sim P_h(x,a)}\left[\min_{a'} \bar{d}(x',a')\right] \\
&= \mathcal{T}_h^\star \bar{d}(x,a).
\end{aligned}
$$

$\square$

**Lemma H.2** (Performance Difference Lemma (PDL)). *For any $f : (\mathcal{X} \times \mathcal{A} \to \mathbb{R})^H$ and policies $\pi, \pi'$, we have*

$$V^\pi - \mathbb{E}_{a \sim \pi'(x_1)}[f_1(x_1, a)] = \sum_{h=1}^{H} \mathbb{E}_\pi \left[ \mathcal{T}_h^{\pi'} f_{h+1}(x_h, a_h) - f_h(x_h, \pi') \right]. \tag{9}$$

*Proof.* We proceed by inducting on the following claim: for all $h = H+1, H, \ldots, 1$,

$$V_h^\pi(x_h) - f_h(x_h, \pi') = \sum_{t=h}^{H} \mathbb{E}_{\pi, x_h} \left[ \mathcal{T}_t^{\pi'} f_{t+1}(x_t, a_t) - f_t(x_t, \pi') \right].$$

The base case of $H+1$ is trivially true as everything is 0. Now fix any $h$ and suppose the IH at $h+1$ is true. Then

$$
\begin{aligned}
&V_h^\pi(x_h) - f_h(x_h, \pi') \\
&= \mathbb{E}_{\pi, x_h}\left[c_h + V_{h+1}^\pi(x_{h+1}) - f_{h+1}(x_{h+1}, \pi') + f_{h+1}(x_{h+1}, \pi') - f_h(x_h, \pi')\right] \\
&= \mathbb{E}_{\pi, x_h}\left[V_{h+1}^\pi(x_{h+1}) - f_{h+1}(x_{h+1}, \pi')\right] + \mathbb{E}_{\pi, x_h}\left[c_h + f_{h+1}(x_{h+1}, \pi') - f_h(x_h, \pi')\right].
\end{aligned}
$$

By the IH, the first term is equal to $\sum_{t=h+1}^{H} \mathbb{E}_{\pi, x_h}\left[\mathcal{T}_t^{\pi'} f_{t+1}(x_t, a_t) - f_t(x_t, \pi')\right]$. The second term is exactly $\mathbb{E}_{\pi, x_h}\left[\mathcal{T}_h^{\pi'} f_{h+1}(x_h, a_h) - f_h(x_h, \pi')\right]$, which concludes the proof. $\square$

## H.2 Proof of Small-Loss Regret and PAC Bounds

Recall that we defined the function class and distribution class, for each $h$, as

$$\mathcal{D}_h(\Pi) = \{(x,a) \mapsto d_h^\pi(x,a) : \pi \in \Pi\} \tag{10}$$

$$\Psi_h = \{(x,a) \mapsto D_\triangle(f(x,a) \parallel \mathcal{T}^{\star,D} f(x,a)) : f \in \mathcal{F}\}.$$

Also, define the '$V$-type' analogs as follows, which will be useful for PAC instead of regret bounds.

$$\mathcal{D}_{h,v}(\Pi) = \{x \mapsto d_h^\pi(x) : \pi \in \Pi\} \tag{11}$$

$$\Psi_{h,v} = \{x \mapsto \mathbb{E}_{a \sim \mathrm{Unif}(\mathcal{A})}[D_\triangle(f(x,a) \parallel \mathcal{T}^{\star,D} f(x,a))] : f \in \mathcal{F}\}.$$

Let us also overload notation for the eluder dimensions as

$$\mathrm{DE}_1(\varepsilon) := \max_h \mathrm{DE}_1(\Psi_h, \mathcal{D}_h(\Pi), \varepsilon),$$

$$\mathrm{DE}_{1,v}(\varepsilon) := \max_h \mathrm{DE}_1(\Psi_{h,v}, \mathcal{D}_{h,v}(\Pi), \varepsilon).$$

Before we prove the following main theorem, a couple of remarks are in order:

1. Recall that by Theorem G.6, we have $\mathrm{DE}_1(\varepsilon) \leq \mathcal{O}(SA \log(SA/\varepsilon))$ and by Theorem G.4, we have $\mathrm{DE}_{1,v}(\varepsilon) \leq \mathcal{O}(d \log(d/\varepsilon))$. This shows that the Eluder dimension in terms in Theorem 5.5 are appropriately bounded.

2. In Appendix D, we showed that distributional BC (Assumption 5.1) is satisfied in low-rank MDPs and the log bracketing number is bounded by $\mathcal{O}(dM \log(d/\varepsilon) + \log|\Phi|)$ where $\Phi$ is a realizable class for $\phi^\star$. This shows that the BC assumption of Theorem 5.5 is satisfied and $\beta$ is appropriately bounded for low-rank MDPs.

Taken together, these two points imply that we have a small-loss PAC bound for low-rank MDPs: concretely, we have $V^{\bar\pi} - V^\star \leq \widetilde{\mathcal{O}}\left(dH\sqrt{\frac{AV^\star \log|\Phi|}{K}} + \frac{d^2 H^2 A \log|\Phi|}{K}\right)$.

We now prove the our main result for online RL: Theorem 5.5. We will prove the result with general function classes, so we will replace the $|\mathcal{F}|$ by its $\ell_\infty$ bracketing number, *i.e.*, $\beta = \log(HKN_{[]}(1/K, \mathcal{F}, \ell_\infty)/\delta)$.

**Theorem 5.5.** *Suppose DistBC holds (Assumption 5.1). For any $\delta \in (0,1)$, w.p. at least $1 - \delta$, running* O-DISCO *with $\beta = \log(HK|\mathcal{F}|/\delta)$ guarantees the following regret bound,*

$$\mathrm{Regret}_{\text{O-DISCO}}(K) \leq 160H\sqrt{KV^\star \mathrm{DE}_1(1/K)\log(K)\beta} + 18000H^2 \mathrm{DE}_1(1/K)\log(K)\beta.$$

*If* UAE $=$ TRUE *(Algorithm 4), then the learned mixture policy $\bar\pi$ is guaranteed to satisfy,*

$$V^{\bar\pi} - V^\star \leq 160H\sqrt{\frac{AV^\star \mathrm{DE}_{1,v}(1/K)\log(K)\beta}{K}} + \frac{18000H^2 A \mathrm{DE}_{1,v}(1/K)\log(K)\beta}{K}.$$

*Proof.* For shorthand, let $\delta_{h,k}(x,a) := D_\triangle(f_h^{(k)}(x,a) \parallel \mathcal{T}_h^{\star,D} f_{h+1}^{(k)}(x,a))$ and $\Delta_k := \sum_{h=1}^H \mathbb{E}_{\pi^k}[\delta_{h,k}(x_h, a_h)]$. Notice that since $\pi_{h+1}^k(x) = \arg\min_a \bar{f}_{h+1}^{(k)}(x,a)$, we have $\mathcal{T}_h^{\pi^k,D} f_{h+1}^{(k)}(x,a) = \mathcal{T}_h^{\star,D} f_{h+1}^{(k)}(x,a)$, so $\delta_{h,k}(x,a) = D_\triangle(f_h^{(k)}(x,a) \parallel \mathcal{T}_h^{\pi^k,D} f_{h+1}^{(k)}(x,a))$ as well.

By Theorem F.2, we have the following two facts **for all** $k \in [K]$,
(i) Optimism: $\min_a \bar{f}_1^{(k)}(x_1, a) \leq V^\star$ (since $Z^\star \in \mathcal{F}_k$) and
(ii) Low training error: for all $h$, we have

If UAE=FALSE. $\sum_{i<k} \mathbb{E}_{\pi^i}[\delta_{h,k}(s_h, a_h)] \leq 240\beta$.

If UAE=TRUE. $\sum_{i<k} \mathbb{E}_{\pi^i}\left[\mathbb{E}_{a' \sim \mathrm{unif}(\mathcal{A})}[\delta_{h,k}(s_h, a_h)]\right] \leq 240\beta$.

The 240 comes from the constants of Theorem F.2 and the fact that $D_\triangle(a,b) \leq 4H^2(a,b)$ for all distributions $a, b$.

Now, fix any episode $k \in [K]$.

$$V^{\pi^k} - V^\star$$

$$\leq V^{\pi^k} - \min_a \bar{f}_1^{(k)}(x_1, a) \qquad \text{(Fact (i))}$$

$$= \sum_{h=1}^{H} \mathbb{E}_{\pi^k} \left[ \mathcal{T}_h^{\pi^k} \bar{f}_{h+1}^{(k)}(x_h, a_h) - \bar{f}_h^{(k)}(x_h, \pi_h^k(x_h)) \right] \qquad \text{(PDL Lemma H.2)}$$

$$= \sum_{h=1}^{H} \mathbb{E}_{\pi^k} \left[ \mathcal{T}_h^{\pi^k, D} f_{h+1}^{(k)}(x_h, a_h) - \bar{f}_h^{(k)}(x_h, a_h) \right] \qquad \text{(Lemma H.1)}$$

$$\leq \sum_{h=1}^{H} \sqrt{\mathbb{E}_{\pi^k} \left[ 4\bar{f}_h^{(k)}(x_h, a_h) + \delta_{h,k}(x_h, a_h) \right]} \cdot \sqrt{\mathbb{E}_{\pi^k} [\delta_{h,k}(x_h, a_h)]} \qquad \text{(Eq. ($\triangle_2$))}$$

$$\leq \sum_{h=1}^{H} \sqrt{4eV^{\pi^k} + 17H \sum_{t=h}^{H} \mathbb{E}_{\pi^k} [\delta_{t,k}(x_t, a_t)]} \cdot \sqrt{\mathbb{E}_{\pi^k} [\delta_{h,k}(x_h, a_h)]}$$

$$\text{(Lemma H.3 and } \mathbb{E}_\pi [Q_h^\pi(s_h, a_h)] \leq V^\pi)$$

$$\leq \sqrt{4eV^{\pi^k} + 17H\Delta_k} \cdot \sqrt{H\Delta_k} \qquad (\star)$$

$$\leq \sqrt{4eHV^{\pi^k}\Delta_k} + 5H\Delta_k$$

$$\leq 2\sqrt{H}\eta^{-1}V^{\pi^k} + 2\sqrt{H}\eta\Delta_k + 5H\Delta_k.$$

In $\star$, we used Cauchy Schwartz. Setting $\eta = 4\sqrt{H}$ and rearranging, we have

$$V^{\pi^k} \leq 2V^\star + 16H\Delta_k + 10H\Delta_k \leq 2V^\star + 26H\Delta_k.$$

Plugging this into $\star$, and noting $104e + 17 \leq 300$, we have

$$V^{\pi^k} - V^\star \leq \sqrt{8eV^\star + 300H\Delta_k}\sqrt{H\Delta_k}.$$

Thus, summing the instantaneous regrets over all episodes, we get

$$\sum_{k=1}^{K} V^{\pi^k} - V^\star \leq \sum_{k=1}^{K} \sqrt{8eV^\star + 300H\Delta_k}\sqrt{H\Delta_k}$$

$$\leq \sqrt{8eKV^\star + 300H\sum_k \Delta_k}\sqrt{H\sum_k \Delta_k} \qquad \text{(Cauchy-Schwartz)}$$

$$\leq 5\sqrt{HKV^\star \sum_k \Delta_k} + 18H\sum_k \Delta_k.$$

**Last step: bounding $\sum_k \Delta_k$.** In this final step, we invoke the pigeonhole property of the eluder dimension, as proven in Theorem 5.3. Note that the precondition of Theorem 5.3 is satisfied by Fact (ii) mentioned at the beginning of this proof. Also, since the triangular discrimination is always bounded by 1, we have that $C$ in Theorem 5.3 is at most 1, and we will also pick $\varepsilon = 1/K$.

On one hand, if UAE=FALSE, then,

$$\sum_{k=1}^{K} \Delta_k = \sum_{h=1}^{H}\sum_{k=1}^{K} \mathbb{E}_{\pi^k}[\delta_{h,k}(x_h, a_h)] \leq 1000H \, \text{DE}_1(1/K)\beta \log(K).$$

On the other hand, if UAE=TRUE, then, we use the V-type analogs,

$$\sum_{k=1}^{K} \Delta_k = \sum_{h=1}^{H}\sum_{k=1}^{K} \mathbb{E}_{\pi^k}[\delta_{h,k}(x_h, a_h)]$$

$$\leq A \sum_{h=1}^{H}\sum_{k=1}^{K} \mathbb{E}_{\pi^k} \left[ \mathbb{E}_{a \sim \text{unif}(\mathcal{A})} \delta_{h,k}(x_h, a) \right]$$

$$\leq 1000AH \, \text{DE}_1(1/K)\beta \log(K).$$

This concludes the proof for both the regret and PAC bounds. □

**Lemma H.3** (Self-bounding lemma). *Let $f \in \mathcal{F}$ and let $\pi$ be any policy. Let us denote $\delta_h(x, a) := D_\triangle(f_h(x, a) \parallel \mathcal{T}_h^{\pi,D} f_{h+1}(x, a))$. Then, for all $h \in [H]$, for all $x_h, a_h$, we have*

$$\bar{f}_h(x_h, a_h) \leq eQ_h^\pi(x_h, a_h) + 4H \sum_{t=h}^{H} \mathbb{E}_{\pi, x_h, a_h}[\delta_t(x_t, a_t)].$$

*Proof.* We prove the following refined subclaim inductively: for all $h \in [H]$, for all $x_h, a_h$, we have

$$\bar{f}_h(x_h, a_h) \leq \sum_{t=h}^{H} \left(1 + \frac{1}{H}\right)^{t-h} \mathbb{E}_{\pi, x_h, a_h}[\bar{c}_t(x_t, a_t) + 2H\delta_t(x_t, a_t)]. \quad \text{(IH)}$$

For $H + 1$ this is trivially true. Now fix any $h$ and suppose IH is true for $h + 1$. By Eq. ($\triangle_2$), for any $h, x_h, a_h$, we have,

$$\bar{f}_h(x_h, a_h) - \mathcal{T}_h^\pi \bar{f}_{h+1}(x_h, a_h) \leq \sqrt{4\mathcal{T}_h^\pi \bar{f}_{h+1}(x_h, a_h) + \delta_h(x_h, a_h)} \sqrt{\delta_h(x_h, a_h)}$$

$$\leq \sqrt{4\mathcal{T}_h^\pi \bar{f}_{h+1}(x_h, a_h)\delta_h(x_h, a_h)} + \delta_h(x_h, a_h)$$

$$\leq \frac{1}{H}\mathcal{T}_h^\pi \bar{f}_{h+1}(x_h, a_h) + (H + 1)\delta_h(x_h, a_h). \quad \text{(AM-GM)}$$

In particular, we have that

$$\bar{f}_h(x_h, a_h)$$
$$\leq \left(1 + \frac{1}{H}\right)\mathcal{T}_h^\pi \bar{f}_{h+1}(x_h, a_h) + 2H\delta_h(x_h, a_h)$$
$$= \left(1 + \frac{1}{H}\right)\left(\bar{c}_h(x_h, a_h) + \mathbb{E}_{x_{h+1} \sim P_h^\star(x_h, a_h)}[\bar{f}_{h+1}(x_{h+1}, \pi)]\right) + 2H\delta_h(x_h, a_h)$$
$$\leq \left(1 + \frac{1}{H}\right)\left(\bar{c}_h(x_h, a_h) + \mathbb{E}_{x_{h+1} \sim P_h^\star(x_h, a_h)}\left[\sum_{t=h+1}^{H} \left(1 + \frac{1}{H}\right)^{t-h-1} \mathbb{E}_{\pi, x_{h+1}}[\bar{c}_t(x_t, a_t) + 2H\delta_t(x_t, a_t)]\right]\right)$$
$$\quad \text{(IH)}$$
$$+ 2H\delta_h(x_h, a_h),$$

which proves the inductive claim. Noting that $\sum_{t=1}^{H}(1 + 1/H)^t \leq e$, we have proven the lemma. □

### H.3 Regret Bounds for Tabular MDPs

**Theorem H.4** (Small-loss regret for tabular MDP). *Suppose the MDP is tabular with $X$ states and assume Assumption 5.1. Fix any $\delta \in (0, 1)$ and set $\beta = \log(HK|\mathcal{F}|/\delta)$. Then, w.p. at least $1 - \delta$,*

$$\text{Regret}_{\text{O-DISCO}}(K) \in \mathcal{O}(H\sqrt{XAKV^\star\beta} + H^2XA\beta).$$

In terms of $H, X, A, K$ scaling, our bound matches that of GOLF [Xie et al., 2023] and is only a $H$ factor looser than that of the minimax lower bound $\widetilde{\mathcal{O}}(\sqrt{XAK})$. The key benefit over prior bounds is that our leading term scales with the minimum cost of the problem $V^\star$. For example, if $V^\star \approx 0$, O-DISCO attains $\mathcal{O}(\log K)$ regret while uniform regret bounds are lower bounded by $\Omega(\sqrt{K})$. Compared to the minimax-optimal UCBVI [Azar et al., 2017], one weakness of our theorem is that it needs a $\mathcal{F}$ satisfying BC. Fortunately, in tabular MDPs where cost is only revealed at the last step from a known distribution, we can choose $\mathcal{F}_{tab}$ as described in Wu et al. [2023, Lemma 4.15] to automatically satisfy BC. By extending our theory via bracketing entropy (Appendix F), we can derive that $\mathcal{F}_{tab}$ yields $\beta = \mathcal{O}(X^2A^2 \log(XAHK/\delta))$. We note that if costs are unknown but discrete, it is possible to construct a BC function class with $\beta$ scaling as $\mathcal{O}(X^2A^2 \log(nXAHK/\delta))$ where $n$ is the maximum number of possible cumulative costs.

**Extension to linear MDPs** The Q-type dimension captures Linear MDPs when squared loss is used by exploiting the fact that the bellman residual is linear in $\phi^\star(x,a)$ [Jin et al., 2021a]. However, since our function class is the set of triangular discriminations, rather than the Bellman residual, we find that the Q-type dimension does not immediately capture Linear MDPs unless regularity assumptions are made. For instance, we believe that Linear MDPs are captured by the Q-type dimension if we assume that $Z_h^\pi(z \mid x, a)$ is lower bounded, *i.e.*, the value distribution is sufficiently smooth.

# I  Proofs for Offline RL

**Theorem 6.1** (Small-Loss PAC bound for P-DISCO). *Assume Assumption 5.1. For any $\delta \in (0,1)$, w.p. at least $1 - \delta$, running P-DISCO with $\beta = \log(H|\Pi||\mathcal{F}|/\delta)$ learns a policy $\widehat{\pi}$ that enjoys the following PAC bound with respect to any comparator policy $\widetilde{\pi} \in \Pi$:*

$$V^{\widehat{\pi}} - V^{\widetilde{\pi}} \leq 9H\sqrt{\frac{C^{\widetilde{\pi}}V^{\widetilde{\pi}}\beta}{N}} + \frac{30H^2 C^{\widetilde{\pi}}\beta}{N}.$$

*Proof of Theorem 6.1.* For shorthand, let $\delta_h^\pi(x,a) = D_\triangle(f_h^\pi(x,a) \parallel \mathcal{T}_h^{\pi,D} f_{h+1}^\pi(x,a))$ and $\Delta^\pi = \sum_{h=1}^H \mathbb{E}_\pi[\delta_h^\pi(x_h,a_h)]$. Also, let $\bar{f}(x,\pi) = \mathbb{E}_{a\sim\pi(x)}[f(x,a)]$.

By Theorem F.3, we have the following two facts, for all $\pi \in \Pi$,
(i) Pessimism: $V^\pi \leq \bar{f}_1^\pi(x_1,\pi)$ (since $Z^\pi \in \mathcal{F}_\pi$) for all $\pi \in \Pi$, and
(ii) $\mathbb{E}_{\nu_h}[\delta_h^\pi(x_h,a_h)] \leq \beta' N^{-1}$ for all $h$ where Theorem F.3 and the fact that $D_\triangle \leq 4H^2$ certifies that $\beta' = 240\beta$ is sufficient.

With these two facts, we can bound the suboptimality of $\widehat{\pi}$ as follows:

$$
\begin{aligned}
&V^{\widehat{\pi}} - V^{\widetilde{\pi}} \\
&\leq \bar{f}_1^{\widehat{\pi}}(x_1,\widehat{\pi}) - V^{\widetilde{\pi}} && \text{(Fact (i))} \\
&\leq \bar{f}_1^{\widetilde{\pi}}(x_1,\widetilde{\pi}) - V^{\widetilde{\pi}} && \text{(Policy selection scheme in Algorithm 3 (Line 4))} \\
&= \sum_{h=1}^H \mathbb{E}_{\widetilde{\pi}}\Big[\bar{f}_h^{\widetilde{\pi}}(x_h,\widetilde{\pi}) - \mathcal{T}_h^{\widetilde{\pi}} \bar{f}_{h+1}^{\widetilde{\pi}}(x_h,a_h)\Big] && \text{(PDL Lemma H.2)} \\
&\leq \sum_{h=1}^H \sqrt{\mathbb{E}_{\widetilde{\pi}}\Big[4\bar{f}_h^{\widetilde{\pi}}(x_h,a_h) + \delta_h^{\widetilde{\pi}}(x_h,a_h)\Big]}\sqrt{\mathbb{E}_{\widetilde{\pi}}\Big[\delta_h^{\widetilde{\pi}}(x_h,a_h)\Big]} && \text{(Eq. ($\triangle_2$))} \\
&\leq \sum_{h=1}^H \sqrt{4eV^{\widetilde{\pi}} + 17H\sum_{t=h}^H \mathbb{E}_{\widetilde{\pi}}\big[\delta_t^{\widetilde{\pi}}(x_t,a_t)\big]}\sqrt{\mathbb{E}_{\widetilde{\pi}}\big[\delta_h^{\widetilde{\pi}}(x_h,a_h)\big]} && \text{(Lemma H.3)} \\
&\leq \sqrt{4eV^{\widetilde{\pi}} + 17H\Delta^{\widetilde{\pi}}}\sqrt{H\Delta^{\widetilde{\pi}}} \\
&\leq 4\sqrt{HV^{\widetilde{\pi}}\Delta^{\widetilde{\pi}}} + 5H\Delta^{\widetilde{\pi}}.
\end{aligned}
$$

Finally, we can bound $\Delta^{\widetilde{\pi}}$ by a change of measure,

$$
\begin{aligned}
\Delta^{\widetilde{\pi}} &= \sum_{h=1}^H \mathbb{E}_{\widetilde{\pi}}\Big[\delta_h^{\widetilde{\pi}}(x_h,a_h)\Big] \\
&\leq C^{\widetilde{\pi}} \sum_{h=1}^H \mathbb{E}_{\nu_h}[\delta_h(x_h,a_h)] \\
&\leq C^{\widetilde{\pi}} H \cdot \beta' N^{-1}. && \text{(Fact (ii))}
\end{aligned}
$$

Therefore,

$$V^{\widehat{\pi}} - V^{\widetilde{\pi}} \leq 4H\sqrt{\frac{C^{\widetilde{\pi}}V^{\widetilde{\pi}}\beta'}{N}} + \frac{5H^2 C^{\widetilde{\pi}}\beta'}{N}.$$

$\square$

# J  Extension: Small-Return Bounds

In this section, we show that O-DISCO and P-DISCO can also be used to obtain small-return bounds. Compared to the algorithms presented in the main text for minimizing cost, we simply have to replace min with max (and vice versa) for maximizing reward, *i.e.*, see Appendix B and enable the SMALLRETURN flag. The proofs are also largely the same, with slight changes to the first few steps.

**Theorem J.1.** *Assume Assumption 5.1 and suppose we want to maximize returns (instead of minimize cost), so enable the* SMALLRETURN *flag. Fix any* $\delta \in (0,1)$ *and set* $\beta = \log(HK|\mathcal{F}|/\delta)$ *and* $\beta' = 60\beta$*. Then, w.p. at least* $1 - \delta$*, running* O-DISCO *(Algorithm 4) with* UAE = FALSE *yields the following small-loss regret bound,*

$$\text{Regret}_{\text{O-DISCO}}(K) \leq 5H\sqrt{KV^\star \text{LSEC}(K)\beta'} + 18H^2 \text{LSEC}(K)\beta'. \tag{12}$$

*If instead* UAE = TRUE*, the outputted policy* $\bar{\pi}$ *enjoys the following small-loss PAC bound,*

$$V^\star - V^{\bar{\pi}} \leq 5H\sqrt{\frac{AV^\star \text{LSEC}_v(K)\beta'}{K}} + 18H^2 \frac{A \text{LSEC}_v(K)\beta'}{K}.$$

*Proof.* Adopt the same notation as in the proof of Theorem 5.5. By Theorem F.2, we have the following two facts for all $k \in [K]$,

(i) Optimism: $V^\star \leq \max_a \bar{f}_1^{(k)}(x_1, a)$ (since $Z^\star \in \mathcal{F}_k$) and

(ii) $\sum_{i<k} \mathbb{E}_{\pi^i}[\delta_{h,k}(s_h, a_h)] \leq \beta'$ for all $h$. If UAE=TRUE, then $a_h$ is sampled from $\text{unif}(\mathcal{A})$ rather than $\pi^i$, *i.e.*, we have $\sum_{i<k} \mathbb{E}_{s_h \sim \pi^i, a_h \sim \text{unif}(\mathcal{A})}[\delta_{h,k}(s_h, a_h)] \leq \beta'$, where $\beta' \lesssim \beta$. Theorem F.2 certifies that $\beta' = 60\beta$ is sufficient.

Fix any episode $k \in [K]$. Then,

$$V^\star - V^{\pi^k}$$

$$\leq \max_a \bar{f}_1^{(k)}(x_1, a) - V^{\pi^k} \tag{Fact (i)}$$

$$= \sum_{h=1}^H \mathbb{E}_{\pi^k}\left[\bar{f}_h^{(k)}(x_h, \pi_h^k(x_h)) - \mathcal{T}_h^{\pi^k} \bar{f}_{h+1}^{(k)}(x_h, a_h)\right] \tag{PDL Lemma H.2}$$

$$= \sum_{h=1}^H \mathbb{E}_{\pi^k}\left[\bar{f}_h^{(k)}(x_h, a_h) - \overline{\mathcal{T}_h^{\pi^k, D} f_{h+1}^{(k)}}(x_h, a_h)\right] \tag{Lemma H.1}$$

$$\leq \sum_{h=1}^H \sqrt{\mathbb{E}_{\pi^k}\left[4\bar{f}_h^{(k)}(x_h, a_h) + \delta_{h,k}(x_h, a_h)\right]} \cdot \sqrt{\mathbb{E}_{\pi^k}[\delta_{h,k}(x_h, a_h)]} \tag{Eq. ($\triangle_2$)}$$

$$\leq \sum_{h=1}^H \sqrt{4eV^{\pi^k} + 17H \sum_{t=h}^H \mathbb{E}_{\pi^k}[\delta_{t,k}(x_t, a_t)]} \cdot \sqrt{\mathbb{E}_{\pi^k}[\delta_{h,k}(x_h, a_h)]}$$

$$\tag{Lemma H.3 and $\mathbb{E}_\pi[Q_h^\pi(s_h, a_h)] \leq V^\pi$}$$

$$\leq \sqrt{4eV^{\pi^k} + 17H\Delta_k} \cdot \sqrt{H\Delta_k} \tag{$\clubsuit$}$$

$$\leq \sqrt{4eV^\star + 17H\Delta_k} \cdot \sqrt{H\Delta_k}$$

Thus, summing the instantaneous regrets over all episodes, we get

$$\sum_{k=1}^K V^{\pi^k} - V^\star \leq \sum_{k=1}^K \sqrt{4eV^\star + 17H\Delta_k}\sqrt{H\Delta_k}$$

$$\leq \sqrt{4eKV^\star + 17H \sum_k \Delta_k}\sqrt{H \sum_k \Delta_k} \tag{Cauchy-Schwartz}$$

$$\leq 5\sqrt{HKV^\star \sum_k \Delta_k} + 18H \sum_k \Delta_k.$$

The bounds for $\Delta_k$ are the same as in Theorem 5.5. $\qquad\square$

In some sense, the proof for the small-returns bound is actually easier than the small-loss bound. Recall that in the cost-minimizing setting, we needed to perform a crucial Cauchy-Schwartz step to rearrange terms at the step labelled ♣. However, in the reward-maximizing setting, we simply bound $V^{\pi^k} \leq V^\star$, without needing to rearrange terms.

**Theorem J.2.** *Assume Assumption 5.1 and suppose we want to maximize returns (instead of minimize cost), so enable the SMALLRETURN flag. Fix any $\delta \in (0,1)$ and set $\beta = \log(H|\Pi||\mathcal{F}|/\delta)$. Then, w.p. at least $1 - \delta$, P-DISCO (Algorithm 4) learns a policy $\widehat{\pi}$ such that for any comparator policy $\widetilde{\pi} \in \Pi$, we have*

$$V^{\widetilde{\pi}} - V^{\widehat{\pi}} \leq 9H\sqrt{\frac{C^{\widetilde{\pi}}V^{\widetilde{\pi}}\beta}{N}} + \frac{30H^2C^{\widetilde{\pi}}\beta}{N}.$$

*Proof of Theorem J.2.* Adopt the same notation as in the proof of Theorem 6.1. By Theorem F.3, we have the following two facts, for all $\pi \in \Pi$,
(i) Pessimism: $\bar{f}_1^\pi(x_1, \pi) \leq V^\pi$ (since $Z^\pi \in \mathcal{F}_\pi$) for all $\pi \in \Pi$, and
(ii) $\mathbb{E}_{\nu_h}[\delta_h^\pi(x_h, a_h)] \leq \beta' N^{-1}$ for all $h$ where $\beta' \leq 60\beta$.

With these two facts, we can bound the suboptimality of $\widehat{\pi}$ as follows:

$$
\begin{aligned}
&V^{\widetilde{\pi}} - V^{\widehat{\pi}} \\
&\leq V^{\widetilde{\pi}} - \bar{f}_1^{\widehat{\pi}}(x_1, \widehat{\pi}) && \text{(Fact (i))} \\
&\leq V^{\widetilde{\pi}} - \bar{f}_1^{\widetilde{\pi}}(x_1, \widetilde{\pi}) && \text{(Policy selection rule in Line 5)} \\
&= \sum_{h=1}^{H} \mathbb{E}_{\widetilde{\pi}}\Big[\mathcal{T}_h^{\widetilde{\pi}}\bar{f}_{h+1}^{\widetilde{\pi}}(x_h, a_h) - \bar{f}_h^{\widetilde{\pi}}(x_h, \widetilde{\pi})\Big] && \text{(PDL Lemma H.2)} \\
&\leq \sum_{h=1}^{H} \sqrt{\mathbb{E}_{\widetilde{\pi}}\big[4\bar{f}_h^{\widetilde{\pi}}(x_h, a_h) + \delta_h^{\widetilde{\pi}}(x_h, a_h)\big]} \sqrt{\mathbb{E}_{\widetilde{\pi}}\big[\delta_h^{\widetilde{\pi}}(x_h, a_h)\big]}. && \text{(Eq. ($\triangle_2$))}
\end{aligned}
$$

From here, the same argument in the proof of Theorem 6.1 finishes the proof. □

## K Experiment Details

**Experiment Settings**

In our experiments, as outlined in Foster and Krishnamurthy [2021], our $\gamma$ learning rate at each time step $t$ is set to $\gamma_t = \gamma_0 t^p$ where $\gamma_0$ and $p$ are hyperparameters. We use batch sizes of 32 samples per episode, and the King County and Prudential experiments run for $5,000$ episodes while the CIFAR-100 experiment runs for $15,000$.

For each dataset, we select the hyperparameter configuration with the best performance for each algorithm. As we report two metrics, performance over the last 100 episodes and over all episodes, we choose the best hyperparameters for each metric as well. While it is often the same hyperparameters that give the best last 100 episodes and all episodes results for a model, that is not always the case. We use the WandB (Weights and Biases) library to run sweeps over hyperparameters.

**Oracles**

For our regression oracles, we use ResNet18 [He et al., 2016], with a modified output layer (so that the output is suited for 100 prediction classes) for CIFAR-100, and a simple 2 hidden-layer neural network for the Prudential Life Insurance and King County Housing datasets. For DISTCB, the oracle's output layer has size $AC$ where $A$ is the number of actions and $C$ is the number of potential costs. This is reshaped so that for each action, there are predictions associated with each potential cost, which then have a softmax function applied to them to represent cost probabilities. For SquareCB and FastCB, the output size is $A$ because there is just a single prediction associated with each action. As per Foster and Krishnamurthy [2021], a sigmoid function is applied to this output layer. All experiments were implemented using PyTorch.

**Datasets**

We now provide an overview table as well as additional details and context to our setups for each dataset. Note that the number of items in each dataset in the table is the count after preprocessing.

| Datasets | | | |
|---|---|---|---|
| Dataset | Items | Number of Actions | Number of Costs |
| CIFAR-100 | 50,000 | 100 | 3 |
| Prudential Life Insurance | 59,381 | 8 | 9 |
| King County Housing | 20,148 | 100 | 101 |

Table 3: Overview of the three datasets and their experimental setups

**Prudential Life Insurance** This dataset is from the Prudential Life Insurance Kaggle competition [Montoya et al., 2015]. It is featured in Farsang et al. [2022], which inspires our experimental setup. The risk level in [8] directly determines the price charged to the customer. Thus, we can consider the chosen risk level as the action taken. If the model overpredicts the risk level, we get a cost of $1.0$ because this is considered over charging the customer and not getting a sale. Otherwise, the model's prediction is charging too little for the customer. To reiterate, the cost in this case is $.1 * (y - \hat{y})$ where $y$ is the actual risk level, and $\hat{y}$ is the predicted risk level.

**King County Housing** The King County housing dataset is also used in Farsang et al. [2022]. An interesting part of the setup is that the cost construction in the case of not overpredicting differs from the Prudential experiment, even though they're both effectively about predicting a price point. Here, the model's chosen price is considered the gain, which is why the cost is $1.0$ minus the chosen price. On the other hand, in the Prudential experiment, the cost is a linear function of the difference between the chosen value and the actual value.

**CIFAR-100** For the CIFAR-100 experiment, we use the training dataset of $50,000$ images as our dataset. The inclusion of the superclass is critical, as it lets us delineate 3 possible costs that DISTCB can learn. Without the super class, the cost construction would be a pure binary of correct vs. incorrect. If this were the case, the ability to test the effectiveness of learning the distribution would be nullified. The distribution would just be whether an action is correct or not, which means our algorithm would essentially be predicting the mean directly.

**Results**

The largest advantages DISTCB had over the next best algorithm were in the Prudential experiment, with DISTCB having a .086 advantage over the last 100 episodes and a .045 advantage over all episodes. While the gaps were not as large for the other two datasets, they are still statistically significant and further showcase the benefit of distribution learning.

