# OpenReview forum: "The Benefits of Being Distributional: Small-Loss Bounds for Reinforcement Learning"
_NeurIPS.cc/2023/Conference — NeurIPS 2023 poster_

### Official Review · Reviewer_8JhK · 2023-07-03

**Soundness:** 3 good
**Presentation:** 4 excellent
**Contribution:** 3 good
**Rating:** 7
**Confidence:** 4

**Summary:**

This paper explores the benefits of distributional RL and provides a theoretical understanding of its advantages through the lens of small-loss bounds. The authors propose distributional algorithms for contextual bandits, online RL, and ofﬂine RL, and prove that these algorithms achieve small-loss regret bounds.


**Strengths:**

I appreciate the attempts to understand the practically powerful algorithm and also the provided empirical simulations, which should be the promising direction in the RL theory study.

The obtained bound scales with the minimum loss so can outperform standard results when it is small. This result seems to be the first line of work in the literature on RL.

The paper is also very well-written and easy to follow.

**Weaknesses:**

1 The algorithm of RL setting still follows 1) reduce the out-of-sample target (immediate regret) to the in-sample loss by complexity measure (SEC); 2) maintain a version space by effective estimation of in-sample loss and choose the optimistic estimator to handle the difference between V^* and V_f (min_a f_1 in this case). The technical novelty is rather limited since similar treatments have been developed in the literature. Also, as far as I know, there are some proofs based on Banach fixed point theorem for the distributional RL, which typically assumes that we can always solve the sub-optimization problems. These results should have supported the effectiveness of the distributional RL to some degree.

2 The proposed algorithm relies on the (distributional) Bellman completeness assumption, which is hard to satisfy because it is non-monotone (adding a new function can violate this assumption). While I understand it is common in the literature to handle the double-sampling issue, it limits the contribution of this work if the goal is to provide a theoretical understanding of the DRL algorithms. I am a little bit confused indeed because, in my mind, the MLE analysis does not require a completeness assumption. Does the distributional setting lead to unique technical challenges here so you have to make such an assumption? In particular, I am wondering whether it is possible to replace it with the trajectory average technique (at a cost of worse regret bound) as in [2,3].

3 The definition of linear SEC. I am wondering whether there are more non-trivial examples captured by it. For linear structure, my understanding is that the square comes from Cauchy-Schwarz to relate $|\phi(z_t)^T \theta_f|$ to $||\phi(z_t)||_{\Sigma_t^{-1}}$

and $ ||\theta_f||_{\Sigma_t} $. The square also exists in the definition of the standard eluder dimension. One exception is the $\ell_1$ eluder dimension developed in [1] (see the end of section 5) where both the out-of-sample prediction error and the in-sample training error are linear. I am wondering whether there is any direct application or corollary between them. Another minor comment is that you may also mention the decoupling/eluder coefficient in [4, 5] where such a reduction treatment is first proposed and generalized by them and you can easily bound the complexity measures via SEC.

**The generality of the framework is indeed my main concern**. Actually, I personally have obtained a very similar first-order result with OMLE. I think we have a similar technical issue that we need to have a result as in line 778. My choice is to define the eluder dimension as in [7] (so, we can call it linear eluder dimension just like linear SEC) but I could not find more interesting instances covered by it.

4 An interesting point is that you choose to decouple the original target with two types of complexity measures. In contextual bandit, the immediate regret is decoupled to another out-of-sample target plus some decoupling cost (that is why we need an online oracle). It turns out that this type of complexity measure can be sub-optimal (especially in model-free cases, see [6] for examples). I am wondering whether this choice is only for a convention or for technical requirements because otherwise, we can definitely apply the eluder techniques to reduce the immediate regret to the in-sample error as in the RL case (that is why we only need an offline oracle that looks back at the k-1 samples collected so far).


[1] When Is Partially Observable Reinforcement Learning Not Scary?

[2] Contextual decision processes with low bellman rank are pac-learnable

[3] Bilinear classes: A structural framework for provable generalization in rl

[4] A Provably Efficient Model-Free Posterior Sampling Method for Episodic Reinforcement Learning

[5] GEC: A Unified Framework for Interactive Decision Making in MDP, POMDP, and Beyond

[6] A note on model-free reinforcement learning with the decision-estimation coefficient

[7] Eluder-based Regret for Stochastic Contextual MDP







**Questions:**

see weakness.




**Limitations:**

yes

---

> ### Author Rebuttal · Authors · 2023-08-10
>
> Thanks so much for your constructive feedbacks and please find our responses below.
>
> 1. **a)** You are absolutely right that our RL algorithms use the global optimism & pessimism ideas from GOLF (Jin et al., 2021a) & Bellman-consistent pessimism (Xie et al., 2021), and that is actually exactly our point: by simply changing squared loss regression to distributional RL via MLE (just like how C51 simply changed DQN's square loss regression to MLE) we can obtain much faster finite-sample rates in both online and offline RL. Notably, by only changing the loss from square loss to distributional loss, our theoretical _ablation_ proves that distributional RL is indeed responsible for the faster small-loss bounds. Additionally, we highlight that most of our RL results are the first small-loss bounds in these settings (e.g. offline RL and online non-tabular RL), which we believe to be a significant novel contribution in its own right. \
> **b)** Regarding prior distributional RL (DRL) results with Banach fixed point, these convergence results are *asymptotic* to the best of our knowledge, and hence do not show any *benefits* of DRL compared to vanilla non-distributional RL, i.e. DRL's bounds are not any better than those from vanilla RL. Our key novelty is that our small-loss bounds have a *faster convergence rate* ($\widetilde{\mathcal{O}}(1/N)$ in small-loss regime) than the typical $\widetilde{\Omega}(1/\sqrt{N})$ rates attainable with non-distributional methods; in this sense, our work is the first to illustrate concrete theoretical *benefits* of DRL compared to vanilla non-distributional approaches.
> 2. **a)** You are correct that MLE by itself does not require BC (see eg. Theorem E.4), but since we perform MLE recursively in a TD-fashion (i.e. the target of the MLE depends on the learned distribution from the last step) we need BC to guarantee all the MLEs will succeed (see eg. Theorem F.2). In other words, the reason BC appears here is exactly why it also appears in the non-distributional setting: squared loss regression itself does not require BC, but TD-style methods like GOLF and FQI do require BC to guarantee squared loss regression succeeds each time. \
> **b)** Please also see our discussion on BC in the global response. We can prove distributional BC for tabular, linear, low-rank, and LQR MDPs. We will add this to the paper to address your important point.
> 3. Thank you so much for pointing us in the direction of the $\ell\_1$-eluder dim [1], which inspired a new result we can now prove. The LSEC is indeed bounded by the $\ell_1$ *distributional* eluder dimension in a manner similar to how SEC is bounded by $\ell_2$ distributional eluder dimension, cf. Proposition 7 of the SEC paper. We can also show that $\ell_1$ distributional eluder is bounded by $\ell_2$ distributional eluder, cf. Proposition 19 of [1]. Altogether, we've shown the following chain: $\text{SEC}(\{f^2:f\in\Psi\},\mathcal{D},K)\leq \text{LSEC}(\Psi,\mathcal{D},K)\leq \text{dim}\_{\ell_1}(\Psi,\mathcal{D},1/K)\leq \text{dim}\_{\ell_2}(\Psi,\mathcal{D},1/K),$ where $\text{dim}\_{\ell\_p}$ is the $\ell_p$ distributional eluder dimension.
> It turns out for low-rank MDPs, we can show $\text{dim}_{\ell_2}(\Psi,\mathcal{D},\epsilon)=\mathcal{O}(d\log(1/\epsilon))$. The intuition is to leverage the linear transitions by using the one-step back trick (Lemma 12 of Uehara et al., 2021 "Representation learning for…") and then apply the elliptical potential lemma (Lemmas 19 and 20 of Uehara et al., 2021). Therefore, we've shown that the V-type LSEC (for any function class) is bounded by $\mathcal{O}(d\log(d/\epsilon))$ in low-rank MDPs. Therefore, our framework can prove the first small-loss PAC bound for low-rank MDPs (more generally, any model with low Bellman eluder dimension!), which is significantly more general than latent variable models. We'd like to thank you again for your insightful suggestions. We hope this, along with the global response on BC, clears up any concerns about generality.
> 4. **a)** As you pointed out in [6], model-free DEC can be sub-optimal in RL. However, in CBs, the DEC-like approach we took is actually rate-optimal, as shown by Section 3 of Foster and Rakhlin, 2020 “Beyond UCB: …”.  \
> **b)** As you mentioned, another route is to use optimism with Eluder techniques, which allows for offline oracles. However, this typically has two drawbacks. First, it is more restrictive since the examples covered by it are typically linear. Second, for general function classes, the optimism step is typically computationally hard, while our CB alg is easily implementable (hence, we could provide experimental results on complex, real-world CBs).  \
> **c)** Finally, we want to highlight that this Eluder route is captured as a special case of our online RL algorithm when $H=1$. In this one-step case, the BC assumption is not needed since there is no TD. Using our new result in (3), we can obtain a regret bound with the Eluder dimension. In sum, our algorithms in this paper can capture both the DEC and the Eluder route for CBs.
>
> Thank you again for all your constructive feedback. We will incorporate all these comments in the camera-ready, and please let us know if you have any additional questions.

---

> > ### Comment · Reviewer_8JhK · 2023-08-10
> > **thanks for the response**
> >
> > Congratulations on your nice work. My major concerns have been addressed.
> >
> > I raise my score to 7 to support this paper for acceptance.
> >
> > Another minor comment is that you may check the techniques in [1] to see whether we can find more interesting examples as they also consider the linear eluder dimension of Hellinger/TV.
> >
> > [1] On the Statistical Efficiency of Mean Field Reinforcement Learning with General Function Approximation

---

### Official Review · Reviewer_nAEh · 2023-07-10

**Soundness:** 3 good
**Presentation:** 3 good
**Contribution:** 3 good
**Rating:** 7
**Confidence:** 3

**Summary:**

This paper establishes the regret bounds of the proposed distributional methods from the perspective of the small-loss bound, where the regret upper bound depends not only on the number of iterations but also the optimal value. Based on this, they can show that regret could be lower under those environments whose optimal policy corresponds to a higher value function.

This paper provides three new distribution RL approaches for three different RL settings. The first one combines ReIGW and MLE to solve contextual bandits. The second one and the third one use the similar concept, which apply confidence intervals on the estimation of likelihood, to solve online RL and offline RL, respectively. They show that the proposed method can achieve competitive regret bound theoretically. The numerical results also demonstrate the empirical performance of the distributional approach for contextual bandits.

**Strengths:**

- The concept of small-loss bound is very interesting and appears novel in the context of distributional RL. This new perspective indeed offers a promising way to theoretically understand why distribution methods could empirically achieve better performance in the existing distributional RL literature.
- This paper is thorough in that it considers three important RL settings: (i) contextual bandits, (ii) online RL, and (iii) offline RL. This paper rigorously shows that the distributional MLE methods have provable benefits in terms of regret for a wide variety of RL problems.
- The idea of combining MLE and confidence set is interesting and different from the traditional idea of considering the confidence interval of mean return.

**Weaknesses:**

- Page 7 seems to be missing. This makes the upper half of Page 8 somewhat difficult to read.
- The regret bounds depend on some factors about the size of the distribution class. For example, in contextual bandits, the regret bound depends on Regret_{log}(K). Similarly, in online and offline RL, the regret bounds depend on $|\mathcal{F}|$. The finiteness of distribution classes appears to be a fairly strong assumption in RL.
- Accordingly, another concern is on the assumption that there exists good distribution classes that satisfy Bellman completeness. While it is mentioned in Section 5 that this assumption could hold under some special tabular MDPs, it is unclear how far this argument could go in the more general RL settings.
- The numerical simulations only discuss the result of contextual bandits. While typically I would not complain about the simulations in a theory paper, I do think that experiments on both online RL and offline RL could be very helpful in understanding the connection between small loss bounds and the empirical regrets.

**Questions:**

I agree with the authors that there is very limited theoretical understanding of distributional RL, and therefore overall I can appreciate the contribution of this paper which offers a new viewpoint for understanding distributional methods in RL. That said, the algorithms proposed and analyzed in this paper are all of MLE style, and somehow they are quite distant from the mainstream distributional RL methods (e.g., C51, QR-DQN, IQN, etc). While the results in this paper are nice to have, I am not sure by how much these analytical insights could benefit the understanding of these popular distributional RL methods. In other words, one of my concerns is that the usefulness of small-loss bounds is tied to the specific algorithms presented in this paper. It would be very helpful if the authors could comment on the connection between the small-loss bounds and other common distributional methods.

Some detailed questions:
- Line 55: The paper claims that triangular discrimination is a novel approach for decomposing the regret, but this idea has been adapted from [Foster and Krishnamurthy, 2021.
- Line 131: The cost distribution should be some distribution in [0, H-h]? Similar issue occurs for the distribution of the loss-to-go in Line 146.
- Lines 127&148: The notation \bar{C} and \bar{Z} is not defined.
- Line 152: How to define the sum of two distributions (or two random variables)? Do we assume independence here?
- In Algorithm, line 4 and Algorithm 3, line 3: What is $\bar{f}_1$ in line 3?
- I am not sure how to compute the inner max of the confidence set. It seems that if the policy is not given, the \mathcal{F}, which is a function of {\pi}, is hard to compute.
- There are many hyperlinks pointing to the wrong positions.
- Line 626, the notation in this inequality is misleading, does f_1 only depend on x or (x,y)?

I am starting with 5 and would be willing to raise the score if the authors could address these questions and the issues mentioned above.

**Limitations:**

The authors do not describe any specific limitation of this paper.

---

> ### Author Rebuttal · Authors · 2023-08-10
>
> Thanks so much for your constructive feedback. Please find our responses below.
>
> ### For the Weaknesses section:
> 1. We used the allowed 1 page PDF to upload the contents of Pg. 7, which got accidentally cut. We sincerely apologize for the inconvenience. Luckily no crucial material was accidentally omitted.
> 2. **a)** Depending on $\text{Regret}_{\log}(K)$ for CB regret is fairly standard, e.g. see Theorem 1 in Foster and Rakhlin, 2020 “Beyond UCB: …” and Foster and Krishnamurthy, 2021 ”Efficient first-order…”. Essentially Theorem 4.1 translates the decision-making regret to the regret of online learning of the distribution, which can be bounded for particular classes and learning algorithms. The crucial point is the dependence on $C^\star$. \
> **b)** We consider infinite distribution classes in Appendix F. We mention this at the bottom of page 6 but can certainly advertise this extension much better in the main text. We focused on finite classes in the main text to keep it brief and since finite classes are commonly considered in RL theory. Nonethless, we did extend to infinite classes, as we agree it is much more realistic. The complexity measure we use for infinite classes is the bracketing entropy as it is the standard complexity measure for MLE, cf. Van de Geer, 2000 "Empirical Processes in M-estimation". For example, if $\mathcal{F}$ is a linear function class with features of dimension $d$, then its bracketing entropy is $\mathcal{O}(d)$.
> 3. Please see our discussion on BC in the global response. In short, we can prove distributional BC for tabular, linear, low-rank, and LQR MDPs. We will add this to the paper to address your important point. Thanks for urging us in this direction.
> 4. As you write, our primary contributions are theoretical. Running additional experiments for the RL case may be difficult as our DistRL algorithms are based on GOLF and Bellman-consistent pessimism (BCP), which are version space methods that are NP-hard to run. With that said, we'd like to highlight two possible directions for practical versions of the algorithm. First, our confidence set construction is for deep exploration; if the problem only needs shallow exploration, we can adopt a cheaper exploration strategy such as $\epsilon$-greedy, cf. Dann et al., 2022 "Guarantees for $\epsilon$-greedy…" Second, a follow-up to BCP successfully implemented its main algorithmic idea and showed state-of-the-art results in offline RL benchmarks (Cheng et al., 2022 "Adversarially trained…"). Since our offline RL alg shares similarity with BCP, we believe our work provides strong support for empirically investigating a distributional version of Cheng et al., 2022. We leave the implementation and benchmarking as promising future work.
>
>
> ### For Questions section:
> **Mainstream DistRL algorithms:**
> 1. C51 is actually very similar to MLE, modulo a projection step that is needed due to discreting values. If $Z_{\tilde{\theta}}$ is the learned distribution from the last step, C51's update aims to learn $Z_{\theta}$ that minimizes $KL(\Phi \mathcal{T} Z_{\tilde{\theta}}|| Z_\theta)$, where $\Phi\mathcal{T}$ is the projected distributional Bellman operator. Since minimizing KL is equivalent to MLE, C51 is essentially doing MLE with projection, so our insights may well apply to C51-style methods.
> 2. Quantile-regression (QR) methods such as QR-DQN & IQN minimize the pinball loss rather than maximizing log-likelihood. While we use guarantees in the squared Hellinger from MLE, QR gives guarantees in the Wasserstein distance. It is interesting future work to explore the theoretical benefits of QR for decision making.
>
>
> **Detailed Questions:**
> * Line 55: Foster and Krishnamurthy, 2021 study solely CBs and their analyses do not immediately lead to RL bounds. In contrast, our novel techniques (e.g. self-bounding Lemma G.4) enable us to go beyond CB and prove to the first small-loss bounds in offline and online RL (in non-tabular settings). (Our CB results are not significant in view of Foster and Krishnamurthy and are only meant as expositional warm-up for our RL results, which _are_ novel and significant.)
> * Line 131: The costs-to-go are in [0,1] rather than [0,H-h] since we work under the normalized cumulative costs setup, i.e., costs and cumulative costs are normalized in [0,1] as in Jiang and Agarwal, 2018 “Open Problem: The Dependence…”. This setup allows for sparse costs and is a more general than assuming costs to be normalized in [0,1] up to rescaling by H, i.e. in our setup, 100% of the total cost can be obtained at a single step, while in the traditional setup, only 1/H-fraction of the total cost can be obtained each step.
> * Line 127 & 148: The \bar notation, which we defined in Line 128, denotes averaging over a distribution. We will remind the reader in a few spots.
> * Line 152: Yes; by + we meant convolution here, that is, the distribution of the sum of independent draws from each distribution. We will simply avoid this notation and describe the distribution explicitly. Note this is the standard distributional Bellman operator, cf. Definition 4.8 from Bellemare et al., 2023 "Distributional Reinforcement Learning".
> * Our online RL confidence set does not depend on policies (it is defined wrt Bellman optimality operator). Our offline RL confidence set *does* depend on policies (it is defined wrt policy's Bellman operator). We discuss their computational complexity in the uploaded PDF (Pg 7).
> * We will ensure hyperlinks are fixed for the final version. (Splitting main text and supplement broke all links.)
> * Line 626: Thanks for catching. Where we write "$f_1(x,y)$" on the rhs we meant the density of the distribution $f_1(x)$ evaluated at $y$. We will add the assumption that the density exists and give it a notation.
>
> Thanks again for your constructive feedback. We will incorporate all these comments in the camera-ready, and please let us know if you have any additional questions.

---

> > ### Comment · Reviewer_nAEh · 2023-08-21
> > **Thanks for the response**
> >
> > Thank the authors for the detailed response. My main concerns about the BC condition, the distribution class, and the connection between the mainstream distributional RL and this paper have been addressed. With that said, I raise the score to 7 and vote for acceptance.

---

### Official Review · Reviewer_L483 · 2023-07-11

**Soundness:** 4 excellent
**Presentation:** 2 fair
**Contribution:** 3 good
**Rating:** 6
**Confidence:** 3

**Summary:**

This paper explores the benefits of distributional reinforcement learning (RL) and provides a mathematical basis for its advantages. Traditional RL approaches focus on learning the mean loss-to-go, but recent developments have shown that learning the entire loss distribution can lead to improved performance in various tasks. However, the theoretical understanding of why and when distributional RL works well has been limited. The paper introduces the concept of small-loss bounds, which are instance-dependent bounds based on the minimum achievable cost in the problem. By optimizing over distributional confidence sets constructed through distributional Bellman equations, the proposed algorithms achieve small-loss regret bounds in tabular Markov decision processes (MDPs) and small-loss PAC bounds in latent variable models. The paper also presents a distributional contextual bandit algorithm and an offline RL algorithm with a novel robustness property. Empirical results demonstrate the effectiveness of the distributional RL algorithms in challenging benchmark tasks.

**Strengths:**

This paper investigates an important problem in theoretical RL and deepens our understanding of its benefits based on rigorous mathematical analysis. It proposes new distributional algorithms for contextual bandits, online and offline RL settings and provides corresponding small-loss bounds. The work presented in this paper is novel and original, to the best of my knowledge.

The paper is well-structured, building from the simple setting of contextual bandits and then extending the analysis to more complex RL setups. The paper clearly states the assumptions, theorems and the proof sketches in each section and provides formal algorithms wherever required.

**Weaknesses:**

The readability of the paper is poor owing to the large amount of text/math. It seems the authors have modified the spacing between lines and headings in some parts of the paper in order to adhere to the page limit.

**Questions:**

None.

**Limitations:**

The paper includes a brief discussion of the limitations.

---

> ### Author Rebuttal · Authors · 2023-08-10
>
> Thanks for your positive comments! Distributional RL is indeed quite notation heavy, and since we show its benefits in all three settings of CBs, online RL, and offline RL, we necessarily need to use notations from all three settings. We hope that Appendix A's table of notations can serve as a convenient index for searching notations, and we'll be sure to add more text descriptions to improve readability.

---

### Official Review · Reviewer_9rQf · 2023-07-12

**Soundness:** 3 good
**Presentation:** 3 good
**Contribution:** 3 good
**Rating:** 7
**Confidence:** 2

**Summary:**

The paper's main concern is to theoretically understand why distributional RL achieves good performance. They consider three different settings: contextual bandit, online RL with an optimistic algorithm, and offline RL with a pessimistic algorithm.

In each case, they use MLE to learn a distribution over the unknown cost (bandit case) and loss-to-go (RL case). The key technique that enables the proof is the relating a new notion of distributional regret to the regular regret; this is done by manipulation of distributional divergences. It appears that the key across settings is that distributional divergence gives more fine-grained control over value/cost differences compared to only looking at means.

This enables them to provide small-loss bounds in each case. They validate their findings empirically for the contextual-bandit case.

**Strengths:**

Significance and Originality:

1. The problem considered is important and relevant to practitioners. The authors do a good job in the introduction of motivating the theoretical conundrum.
2. The tools developed by the authors around distributional divergence are novel to the best of my knowledge.

Quality

1. The results appear correct. And the settings considered are comprehensive across contextual bandit, online and online RL.
2. The empirical results display improved performance over well-chosen benchmarks.

**Weaknesses:**

Clarity:

1. The regret decomposition using triangular discrimination, while novel, is intricate and difficult to interpret intuitively. If further intuition were provided, this would help inspire future theoretical work.

2. The writing is notation-heavy and takes a while to parse through and keep track of notation.

3. Relating to point 1, it would be nice if there were further connections to the practical applications around risk-sensitive RL, which are the primary motivating examples for distributional RL. This would help bridge the gap to practitioners.

**Questions:**

1. The triangular discrimination technique seems central to converting distributional divergence into bounds on value difference. Intuitively, what are the main factors that enable the small-loss term to show up when distributional approaches are used? Are there ways to make this term appear without distributional RL? If not, is there reasoning for why it is not possible?

2. How restrictive are the realizability and Bellman completeness assumptions made in the analyses? Do you have a sense of how the techniques could extend to violated assumptions?

3. Would it be easy to generalize the results to linear MDPs? What obstacles may arise?

**Limitations:**

Yes

---

> ### Author Rebuttal · Authors · 2023-08-10
>
> Thanks for your positive and constructive comments! Please find our responses below.
>
> **Triangular discrimination:**
> Intuitively, triangular discrimination bounds give finer control over estimation error than traditional $L_2$ bounds from non-distributional methods. For example, for some target conditional distribution $g(y\mid x)$, imagine we want to estimate its conditional mean $\bar g(\cdot\mid x)$. On one hand, square loss regression would learn a function $\hat f_{sq}:\mathcal{X}\to\mathbb{R}$ such that $|\mathbb{E}\_x[\hat f\_{sq}(x) - \bar g(\cdot\mid x)]|\leq \\\|\hat f\_{sq}(x) - \bar g(\cdot\mid x)\\\|\_{L\_2} = \mathcal{O}(1/\sqrt{N})$. On the other hand, suppose we learn a conditional distribution $\hat f\_{dist}:\mathcal{X}\to\Delta(\mathbb{R})$ with MLE. Using triangular discrimination, we can obtain a *self-bounding* inequality $|\mathbb{E}\_x[\bar{\hat{f}}\_{dist}(\cdot\mid x) - \bar g(\cdot\mid x)]|\leq \sqrt{(\mathbb{E}\_x\bar g(\cdot\mid x) + D\_\triangle(\hat f\_{dist}(\cdot\mid x)||g(\cdot\mid x))) \cdot \mathbb{E}\_xD\_\triangle(\hat f\_{dist}(\cdot\mid x)||g(\cdot\mid x)) }$. This is in fact the *implicit inequality* that can be derived from Eq. $\Delta_1$ on Page 5. By standard MLE generalization results, we expect $\mathbb{E}\_x D\_\triangle(\hat f\_{dist}(\cdot\mid x)||g(\cdot\mid x))=\mathcal{O}(1/N)$. Thus, if $\mathbb{E}_x\bar g(\cdot\mid x)\approx 0$, we expect the bound to converge as $\mathcal{O}(1/N)$, which is faster than squared loss's $\mathcal{O}(1/\sqrt{N})$ rate. This separation between squared loss regression and MLE is actually fundamental, and there already exists a lower bound in the CB setting, see Theorem 2 of Foster et al., 2021 "Efficient first-order contextual bandits..." To summarize, we can only obtain this key self-bounding inequality with MLE, and not squared loss, and this is the key intuition for how we obtain small-loss bounds.
>
> **Realizability and Bellman completeness:**
> Since BC is stronger than realizability, we focus our discussion on BC, which we posted in the global response. In short, we can prove distributional BC for tabular, linear, low-rank, and LQR MDPs. We will add this to the paper to address your important point.
>
> **Generalization to linear MDPs:**
> As remarked in the global response, DistBC indeed covers linear MDPs. Moreover, we proved another new result inspired by Reviewer 8JhK's comments, which shows the LSEC (an analysis tool used in Appendix G.2) is bounded by the Bellman eluder dimension. Since low-rank MDPs have Bellman eluder dimension $\widetilde{\mathcal{O}}(d)$, these two new results imply that our small-loss bounds also hold for low-rank MDPs (and thus also linear MDPs), further generalizing our results!
>
> **Risk-sensitive RL:**
> Risk-sensitive RL is indeed well-motivated for distributional RL, so there is no conundrum there. The long-standing conundrum about distributional RL is regarding the risk-neutral setup: when optimizing expected returns, why can learning the distribution then computing its mean perform better than learning the mean directly? (By Bellman equations, all we need are the expected returns, so why should we learn the distribution?) We answer these questions with small-loss bounds, which *converge faster* than bounds from non-distributional methods; to the best of our knowledge, our work shows the theoretical benefits of distributional RL for the first time. With that said, we believe the techniques developed in our paper could also be useful for deriving small-loss bounds in risk-sensitive settings, and leave that as future work.

---

### Author Rebuttal · Authors · 2023-08-10

We are grateful for all the encouraging and constructive reviews, which have been helpful in improving and polishing our work this week. Amongst all four reviewers, three of them (9rQf, nAEh, 8JhK) inquired about the necessity and generality of distributional Bellman completeness (DistBC), so we'd like to address this in the global response.

**Necessity of BC for TD:**
As remarked in Lines 224-230, BC is necessary for TD-style algorithms to succeed. Without it, TD can diverge or converge to bad fixed points, e.g. Tsitsiklis and Van Roy, 1996 showed such a counterexample. Since our algorithms are distributional versions of GOLF and Bellman-consistent pessimism, which are TD-style algorithms that already require BC, it is quite natural for our results to rely on analogous assumptions to these prior works. One reviewer (8JhK) brought up the issue of non-monotonicity of BC (adding a new function can violate BC): we want to point out our theorems also hold under "generalized BC," a weaker and monotone assumption that there exists function classes $\mathcal{G}\_h$ such that $\mathcal{T}\_h\mathcal{F}\_{h+1}\subseteq \mathcal{G}\_h$ for all $h$ (cf. Assumption 14 of Jin et al., 2021a). If $\mathcal{G}=\mathcal{F}$, this recovers the typical BC assumption. We'll add this as a remark.

In fact, Foster et al., 2022 "Offline RL: Fundamental Barriers..." showed a lower bound that $Q^\star$-realizability and all-policy concentrability (stronger coverage condition than the single-policy one we use!) are not sufficient conditions for sample efficient offline RL. This suggests that removing BC is challenging and would require some other assumptions in its place. Seeking alternative conditions to BC is not our goal here. Instead, our contribution is that distributional versions of prior algorithms can yield small-loss bounds, which provides the first theoretical answer for the long-standing conundrum in the RL community: when optimizing expected returns, why can learning the return distribution and only then computing its mean perform better than learning the mean directly, which is all we need for Bellman's equation?



**(New result) DistBC is satisfied by linear and low-rank MDPs:**
Urged by the excellent feedback, we've proven a new result: in linear MDPs, the following linear function class automatically satisfies DistBC, $\mathcal{F}=\{f(z\mid x,a) = \phi^\star(x,a)^\top w(z): w: [0,1]\to B^d(r)\}$, where $\phi^\star(x,a)$ are the linear MDP's features and $B^d(r)$ is the radius-$r$ $\ell_2$-ball in $\mathbb{R}^d$ (with $r$ chosen appropriately for normalization purposes). This result is analogous to the well-known fact that linear MDPs automatically satisfy (vanilla) BC with a similar linear function class, and the proofs are similar. Moreover, this result easily extends to low-rank MDPs (where $\phi^\star$ is unknown) if we let $f(z\mid x,a)=\phi(x,a)^\top w(z)$ with $\phi$ varying in $\Phi$, assuming it is realizable, $\phi^\star\in\Phi$. We want to point out that DistBC also captures the Linear-Quadratic Regulator (LQR), as shown in Section B.2 of Wu et al., 2023 "Distributional Offline Policy Evaluation...". In sum, DistBC captures linear MDPs, low-rank MDPs, and LQRs, so DistBC essentially captures the same interesting models captured by vanilla BC.


**Conclusion about BC:**
To summarize the above, (1) BC is necessary for TD and assumed by GOLF and Bellman-consistent pessimism (the non-distributional analogs of our algs), and (2) DistBC captures all the interesting models captured by (vanilla) BC. We hope these two points clarify our rationale for assuming DistBC and that it is not strong at all (relative to prior works).

Additionally, we have attached a 1 page PDF containing a proof sketch of Theorem 5.2, the definition of low-rank MDP and a discussion on computational complexity, which we hope clears up any missing notations.

---

### Author Response · Authors · 2023-08-17

We thank all our reviewers for their thoughtful and constructive comments. We're encouraged by the fact that there is a consensus about the technical importance and presentation quality of our work. As the discussion period is nearing the end, please let us know if there are remaining or new questions you would like us to address. If you are reassured of the value of our work by the rebuttal & discussion, then we’d greatly appreciate if you could please acknowledge and correspondingly update your scores. Thank you!

---

### Decision · Program_Chairs · 2023-09-21

**Decision:**

Accept (poster)

**Comment:**

All the reviewers are in agreement that this paper provides a strong conceptual and technical contribution to the burgeoning area of distributional RL. Based on the author-reviewer discussion, the authors are encouraged to include a detailed discussion of the distributional Bellman completeness assumption, including why it is necessary and under what conditions it is satisfied. I also suggest that the authors include the proof that it is satisfied for linear and low-rank MDPs if they are able to fit it in the camera-ready version.